# Bridging Multicalibration and Out-of-distribution Generalization Beyond Covariate Shift

**Jiayun Wu**[*]
Depart. of Computer Science & Tech.
Tsinghua University
Beijing, China 100084
wujy22@mails.tsinghua.edu.cn

**Jiashuo Liu**
Depart. of Computer Science & Tech.
Tsinghua University
Beijing, China 100084
liujiashuo77@gmail.com

**Peng Cui**
Key Laboratory of Pervasive Computing, Ministry of Education
Depart. of Computer Science & Tech., Tsinghua University
Beijing, China 100084
cuip@tsinghua.edu.cn

**Zhiwei Steven Wu**
School of Computer Science
Carnegie Mellon University
Pittsburgh, PA 15213
zhiweiw@cs.cmu.edu

## Abstract

We establish a new model-agnostic optimization framework for out-of-distribution generalization via multicalibration, a criterion that ensures a predictor is calibrated across a family of overlapping groups. Multicalibration is shown to be associated with robustness of statistical inference under covariate shift. We further establish a link between multicalibration and robustness for prediction tasks both under and beyond covariate shift. We accomplish this by extending multicalibration to incorporate grouping functions that consider covariates and labels jointly. This leads to an equivalence of the extended multicalibration and invariance, an objective for robust learning in existence of concept shift. We show a linear structure of the grouping function class spanned by density ratios, resulting in a unifying framework for robust learning by designing specific grouping functions. We propose MC-Pseudolabel[2], a post-processing algorithm to achieve both extended multicalibration and out-of-distribution generalization. The algorithm, with lightweight hyperparameters and optimization through a series of supervised regression steps, achieves superior performance on real-world datasets with distribution shift.

## 1 Introduction

We revisit the problem of out-of-distribution generalization and establish new connections with multicalibration [17], a criterion originating from algorithmic fairness. Multicalibration is a strengthening of calibration, which only requires a predictor $f$ to be correct *on average* within each level set:

$$\mathbb{E}[Y - f(X) \mid f(X)] = 0$$

---

[*]This research was conducted as part of a visit to Carnegie Mellon University.

[2]Code available at: https://github.com/IC-hub/MC-Pseudolabel

38th Conference on Neural Information Processing Systems (NeurIPS 2024).

Calibration is a relatively weak property, as it can be satisfied even by the uninformative constant predictor $f(X) = \mathbb{E}[Y]$ that predicts the average outcome. More broadly, calibration provides only a marginal guarantee that does not extend to sub-populations. Multicalibration [17] mitigates this issue by requiring the calibration to hold over a family of (overlapping) subgroups $\mathcal{H}$: for all $h \in \mathcal{H}$,

$$\mathbb{E}[(Y - f(X)) \, h(X) \mid f(X)] = 0$$

Multicalibration is initially studied as measure of subgroup fairness for boolean grouping functions $h$, with $h(X) = 1$ indicating $X$ is a member of group $h$ [17]. Subsequently, Gopalan et al. [14] and Kim et al. [20] adopt a broader class of real-valued grouping functions that can identify sub-populations through reweighting. The formulation of real-valued grouping function has enabled surprising connections between multicalibration and distribution shifts. Prior work [21, 37] studied how distribution shift *affects* the measure of multicalibration, with a focus on covariate shift where the relationship between $X$ and $Y$ remains fixed. Kim et al. [21] show that whenever the set of real-valued grouping functions $\mathcal{H}$ includes the density ratio between the source and target distributions, a multicalibrated predictor with respect to the source remains calibrated in the shifted target distribution.

Our work substantially expands the connections between multicalibration and distribution shifts. At a high level, our results show that robust prediction under distribution shift can actually be *facilitated* by multicalibration. We extend the notion of multicalibration by incorporating grouping functions that simultaneously consider both covariates $X$ and outcomes $Y$. This extension enables us to go beyond covariate shift and account for concept shift, which is prevalent in practice due to spurious correlation, missing variables, or confounding [31].

**Our contributions.** Based on the introduction of joint grouping functions, we establish new connections between our extended multicalibration notion and algorithmic robustness in the general setting of out-of-distribution generalization, where the target distribution to assess the model is different from the source distribution to learn the model.

1. We first revisit the setting of covariate shift and show multicalibration implies Bayes optimality under covariate shift, provided a sufficiently rich class of grouping functions. Then, in the setting of concept shifts, we show the equivalence of multicalibration and invariance [2], a learning objective to search for a Bayes optimal predictor $\mathbb{E}[Y|\Phi(X)]$ under a representation over features $\Phi(X)$, even though $\mathbb{E}[Y|X]$ is different across target distributions. We show correspondence between an invariant representation $\Phi(X)$ and a multicalibrated predictor $\mathbb{E}[Y|\Phi(X)]$, with a grouping function class containing all density ratios of target distributions and the source distribution.

2. As part of our structural analysis of the new multicalibration concept, we investigate the maximal grouping function class that allows for a nontrivial multicalibrated predictor. For traditional covariate-based grouping functions, the Bayes optimal predictor $f(X) = \mathbb{E}[Y|X]$ is always multicalibrated, which is no longer the case for joint grouping functions. We show the maximal grouping function class is a linear space spanned by the density ratio of the target distributions where the predictor is invariant. As a structural characterization of distribution shift, this leads to an efficient parameterization of the grouping functions by linear combination of a spanning set of density ratios. The spanning set can be flexibly designed to incorporates implicit assumptions of various methodologies for robust learning, including multi-environment learning [39] and hard sample learning [29].

3. We devise a post-processing algorithm to multicalibrate predictors and simultaneously producing invariant predictors. As a multicalibration algorithm, we prove its convergence under Gaussian distributions of data and certify multicalibration upon convergence. As a robust learning algorithm, the procedure is plainly supervised regression with respect to models' hypothesis class and grouping function class, introducing an overhead of linear regression. This stands out from heavy optimization techniques for out-of-distribution generalization, such as bi-level optimization [12, 30] and multi-objective learning [1, 2, 24], which typically involves high-order gradients [36]. The algorithm introduces no extra hyperparameters. This simplifies model selection, which is a significant challenge for out-of-distribution generalization since validation is unavailable where the model is deployed [15]. Under the standard model selection protocol of DomainBed [15], the algorithm achieves superior performance to existing methods in real-world datasets with concept shift, including porverty estimation [44], personal income prediction [7] and power consumption [32, 33] prediction.

## 2 Multicalibration and Bayes Optimality under Covariate Shift

### 2.1 Multicalibration with Joint Grouping Functions

We consider prediction tasks where covariates are denoted by a random vector $X \in \mathcal{X}$ and the target by $Y \in \mathcal{Y}$. Lowercase $x, y$ denote the specific values of these random variables. Predictors are defined as real-valued functions $f : \mathcal{X} \to \mathcal{Y}$. Our theoretical analysis focuses on the setting where $\mathcal{Y} = [0, 1]$. In this context, we propose a new definition of $\ell_2$ approximate multicalibration with joint grouping functions.

**Definition 2.1** (Multicalibration with Joint Grouping Functions). *For a probability measure $P(X, Y)$ and a predictor $f$, let $\mathcal{H} \subset \mathbb{R}^{\mathcal{X} \times \mathcal{Y}}$ be a real-valued grouping function class. We say that $f$ is $\alpha$-approximately $\ell_2$ multicalibrated w.r.t. $\mathcal{H}$ and $P$ if for all $h \in \mathcal{H}$:*

$$K_2(f, h, P) = \int \left( \mathbb{E}_P \left[ h(X, Y)(Y - v) \big| f(X) = v \right] \right)^2 dP_{f(X)}(v) \leq \alpha. \tag{1}$$

*$P_{f(X)}(v) = P(f^{-1}(v))$ is the pushforward measure. We say $f$ is $\alpha$-approximately calibrated if $\mathcal{H}$ includes the constant function $h \equiv 1$. We say $f$ is multicalibrated (calibrated) for $\alpha = 0$. If the grouping function is defined on $X$, which implies $h(x, y_1) = h(x, y_2)$ for any $x \in \mathcal{X}$ and $y_1, y_2 \in \mathcal{Y}$, we abbreviate $h(x, \cdot)$ by $h(x)$.*

Our definition generalizes several notions of (multi)calibration in the literature by specific choices of grouping functions. For example, $K_2(f, 1, P)$ recovers the overall calibration error in the case of a constant grouping function $h \equiv 1$. For boolean grouping functions defined on $X$ [17], $K_2(f, h, P)$ computes the calibration error of the subgroup with $h(x) = 1$. For real-valued grouping functions defined on $X$ [14, 20], $K_2(f, h, P)$ evaluates a reweighted calibration error, whose weights $h(x)$ are proportional to the likelihood of a sample belonging to the subgroup. Furthermore, we propose an extended domain of grouping functions defined on covariates and outcomes jointly, for which the Bayes optimal predictor $\mathbb{E}[Y|x]$ may not be multicalibrated, in contrast to all existing multicalibration frameworks with $X$-based grouping functions. Multicalibration with joint groupings thus implies a distinct learning objective from accuracy, which we will characterize as *invariance* in section 3.

**Example 2.2** (Multicalibration Does Not Imply Bayes Optimality). *Consider covariates $X = (X_1, X_2)^T$ and an outcome $Y$ generated by the following structural equations:*

$$Y = X_1 + \epsilon_1.$$
$$X_2 = Y + \epsilon_2.$$

*$X_1$, $\epsilon_1$, $\epsilon_2$ are independent gaussian variables with zero mean, and variances $\mathbb{E}[\epsilon_1^2] = \sigma_1^2$ and $\mathbb{E}[\epsilon_2^2] = \sigma_2^2$. For a singleton grouping function class containing $h(x, y) = y - x_2$, the Bayes optimal predictor $f(x) = \mathbb{E}[Y|x] = \frac{\sigma_2^2}{\sigma_1^2 + \sigma_2^2} x_1 + \frac{\sigma_1^2}{\sigma_1^2 + \sigma_2^2} x_2$ is not multicalibrated because $\mathbb{E}[h(X, Y)(Y - f(X))] = \frac{\sigma_1^2 \sigma_2^2}{\sigma_1^2 + \sigma_2^2} \neq 0$. However, $g(x) = x_1$ is multicalibrated since $\mathbb{E}[h(X, Y)(Y - g(X))|g(X)] = -\mathbb{E}[\epsilon_2 \epsilon_1 | X_1] = 0$.*

While the Bayes optimal predictor is always multicalibrated for covariate-based grouping functions, it may not be multicalibrated for grouping functions that depend on the outcome. In fact, there may be no predictor that achieves multicalibration in such cases. For example, when we consider a grouping function class that includes both $h \equiv 1$ and $h(x, y) = y$, a multicalibrated predictor $f$ satisfies $f(X) = \mathbb{E}[Y|f(X)]$ for $h \equiv 1$, and $\mathbb{E}[Y^2|f(X)] = \mathbb{E}[Yf(X)|f(X)] = (\mathbb{E}[Y|f(X)])^2$ for $h(x, y) = y$. This implies $\text{Var}[Y|f(X)] = 0$, which is impossible for regression with label noise. Therefore, we study the structure of the *maximal grouping function class* that allows for a multicalibrated predictor in section 4.

Most importantly, multicalibration with joint grouping functions is useful for capturing more general distribution shifts. By interpreting multicalibration error as calibration error reweighted by grouping functions, it quantifies the maximal calibration error for all subgroups associated with grouping functions in $\mathcal{H}$. If grouping functions are defined on $X$, only the covariate distribution $P(X)$ distinguishes between subgroups. In contrast, the joint distribution $P(X, Y)$ differentiates subgroups for joint grouping functions. We will discuss multicalibration with covariate-based grouping functions in the next sub-section and joint grouping functions in section 3.

## 2.2 Multicalibration Implies Bayes Optimality under Covariate Shift

**Settings of Out-of-distribution Generalization.** We characterize distribution shift by an *uncertainty set* of absolutely continuous probability measures, denoted by $\mathcal{P}(X, Y)$, where there is an accessible source measure $P_S \in \mathcal{P}$ and unknown target measure $P_T \in \mathcal{P}$. We use *capital* letters such as $P$ to denote a single probability measure and *lowercase* letters such as $p$ to denote its probability density function. A predictor $f$ is learned in the source distribution $P_S$ and assessed in the target distribution $P_T$. Given a loss function $\ell : \mathcal{Y} \times \mathcal{Y} \to \mathbb{R}$, we evaluate the average risk of a predictor $f$ w.r.t. a probability measure $P$, defined by $R_P(f) := \mathbb{E}_P[\ell(f(X), Y)]$. We focus on $\ell(\hat{y}, y) = (\hat{y} - y)^2$ in our theoretical analyses.

In this subsection we focus on grouping functions $h(x)$ defined on covariates. We will prove approximately multicalibrated predictors simultaneous approaches Bayes optimality in each target distribution with covariate shift, bridging the results of Kim et al. [21] and Globus-Harris et al. [13]. To recap, Kim et al. [21] studies multicalibration under covariate shift and shows that a multicalibrated predictor remains calibrated in target distribution for a sufficiently large grouping function class. Further, it is shown that multicalibration predictors remain multicalibrated under covariate shift [21, 37], assuming the grouping function class $\mathcal{H}$ is closed under some transformation by density ratios (Assumption 2.3.1). Second, Globus-Harris et al. [13] shows multicalibration implies Bayes optimal accuracy [13], assuming $\mathcal{H}$ satisfies a weak learning condition (Assumption 2.3.2). Detailed discussion on other related works is deferred to section A in the appendix.

**Assumption 2.3** (Sufficiency of Grouping Function Class (informal, see Assumption F.1))**.**

*1. (Closure under Covariate Shift) For a set of probability measures $\mathcal{P}(X)$ containing the source measure $P_S(X)$, $h \in \mathcal{H}$ implies $p/p_S \cdot h \in \mathcal{H}$ for any density function $p$ of distributions in $\mathcal{P}$.*

*2. ($(\gamma, \rho)$-Weak Learning Condition) For any $P \in \mathcal{P}(X)P_S(Y|X) \equiv \{P'(X)P_S(Y \mid X) : P' \in \mathcal{P}\}$ with the source measure $P_S(Y|X)$, and every subset $G \subset \mathcal{X}$ with $P(X \in G) > \rho$, if the Bayes optimal predictor $\mathbb{E}_P[Y|X]$ has lower risk than the constant predictor $\mathbb{E}_P[Y|X \in G]$ by a margin $\gamma$, there exists a predictor $h \in \mathcal{H}$ that is also better than the constant predictor with the margin $\gamma$.*

**Theorem 2.4** (Risk Bound under Covariate Shift)**.** *For a source measure $P_S(X, Y)$ and a set of probability measures $\mathcal{P}(X)$ containing $P_S(X)$, given a predictor $f : \mathcal{X} \to [0, 1]$ with finite range $m := |Range(f)|$, consider a grouping function class $\mathcal{H}$ closed under affine transformation and satisfying Assumption 2.3 with $\rho = \gamma/m$. If $f$ is $\frac{\gamma^6}{256m^2}$-approximately $\ell_2$ multicalibrated w.r.t $P_S$ and $\mathcal{H}_1 := \left\{ h \in \mathcal{H} : \max_{x \in \mathcal{X}} h(x)^2 \leq 1 \right\}$, then for any target measure $P_T \in \mathcal{P}(X)P_S(Y|X)$,*

$$R_{P_T}(f) \leq \inf_{f^* : \mathcal{X} \to [0, 1]} R_{P_T}(f^*) + 3\gamma. \tag{2}$$

**Remark 2.5.** *Following prior work in multicalibration [13, 37], we study functions $f$ with finite cardinality, which can be obtained by discretization.*

## 3 Multicalibration and Invariance under Concept Shift

Theorem 2.4 shows multicalibration implies Bayes optimal accuracy for target distributions under covariate shift. However, in practical scenarios, there are both marginal distribution shifts of covariates ($X$) and *concept shift* of the conditional distributions ($Y|X$). Concept shift is especially prevalent in tabular data due to missing variables and confounding [31]. In order to go beyond covariate shift, we will focus on grouping functions defined on covariates and outcomes jointly. We show that multicalibration notion w.r.t. joint grouping functions is equivalent to invariance, a criterion for robust prediction under concept shift. Extending the robustness of multicalibration to general shift is non-trivial. The fundamental challenge is that there is no shared predictor that is generally optimal in each target distribution because the Bayes optimal predictor varies for different $Y|X$ distributions. As a first step, we show multicalibrated predictors w.r.t. joint grouping functions are robust as they are optimal over any post-processing functions in each target distribution.

**Theorem 3.1** (Risk Bound under Concept Shift)**.** *For a set of absolutely continuous probability measures $\mathcal{P}(X, Y)$ containing the source measure $P_S(X, Y)$, consider a predictor $f : \mathcal{X} \to [0, 1]$. Assume the grouping function class $\mathcal{H}$ satisfies the following condition:*

$$\mathcal{H} \supset \left\{ h(x, y) = \frac{p(x, y)}{p_S(x, y)} \Big| P \in \mathcal{P}(X, Y) \right\}. \tag{3}$$

*If $f$ is $\alpha$-approximately $\ell_2$ multicalibrated w.r.t. $\mathcal{H}$ and $P_S$, then for any measure $P \in \mathcal{P}(X, Y)$,*

$$R_P(f) \leq \inf_{g:[0,1]\to[0,1]} R_P(g \circ f) + 2\sqrt{\alpha}. \tag{4}$$

The theorem shows an *approximately multicalibrated* predictor on the source *almost cannot be improved by post-processing* for each target distribution. To ensure such robustness, the grouping function class must include all density ratios between target and source measures, which are functions over $\mathcal{X} \times \mathcal{Y}$. This characterization of robustness in terms of post-processing echoes with Invariant Risk Minimization (IRM) [2], a paradigm for out-of-distribution generalization with $Y|X$ shift. However, their analysis focuses on representation learning.

**Definition 3.2** (Invariant Predictor). *Consider data selected from multiple environments in the set $\mathscr{E}$ where the probability measure in an environment $e \in \mathscr{E}$ is denoted by $P_e(X, Y)$. Denote the representation over covariates by a measurable function $\Phi(x)$. We say that $\Phi$ elicits an $\alpha$-approximately invariant predictor $g^* \circ \Phi$ across $\mathscr{E}$ if there exists a function $g^* \in \mathcal{G} := \{g : supp(\Phi) \to [0,1]\}$ such that for all $e \in \mathscr{E}$:*

$$R_{P_e}(g^* \circ \Phi) \leq \inf_{g \in \mathcal{G}} R_{P_e}(g \circ \Phi) + \alpha. \tag{5}$$

**Remark 3.3.** *(1) Predictors in $\mathcal{G}$ take a representation $\Phi$ extracted from the covariates as input. For a general predictor $f(x)$, if we take $\Phi(x) = f(x)$ and $g^*$ as an identity function, Equation 5 reduces to the form of Equation 4. Therefore, $f$ in Equation 4 is a $2\sqrt{\alpha}$-approximately invariant predictor across environments collected from the uncertainty set $\mathcal{P}$. (2) We give an approximate definition of invariant predictors, which recovers the original definition [2] when $\alpha = 0$. In this case, there exists a shared Bayes optimal predictor $g^\star$ across environments, taking $\Phi$ as input. This implies $\mathbb{E}_{e_1}[Y|\Phi] = \mathbb{E}_{e_2}[Y|\Phi]$ almost surely for any $e_1, e_2$.*

IRM searches for a representation such that the optimal predictors upon the representation are *invariant* across environments. Motivated from causality, the interaction between outcomes and their causes are also assumed invariant, so IRM learns a representation of causal variables for stable prediction. We extend Theorem 3.1 to representation learning and prove equivalence between multicalibrated and invariant predictors.

**Theorem 3.4** (Equivalence of Multicalibration and Invariance). *Assume samples are drawn from an environment $e \in \mathscr{E}$ with a prior $P_S(e)$ such that $\sum_{e \in \mathscr{E}} P_S(e) = 1$ and $P_S(e) > 0$. The overall population satisfies $P_S(X, Y) = \sum_{e \in \mathscr{E}} P_e(X, Y)P_S(e)$ where $P_e(X, Y)$ is the environment-specific absolutely continuous measure. With a measurable function $\Phi(x)$, define a function class $\mathcal{H}$ as:*

$$\mathcal{H} := \left\{ h(x, y) = \frac{p_e(x, y)}{p_S(x, y)} \Big| e \in \mathscr{E} \right\}. \tag{6}$$

*1. If there is a bijection $g^\star : supp(\Phi) \to [0, 1]$ such that $g^\star \circ \Phi$ is $\alpha$-approximately $\ell_2$ multicalibrated w.r.t. $\mathcal{H}$ and $P_S$, then $\Phi$ elicits an $2\sqrt{\alpha}$-approximately invariant predictor $g^\star \circ \Phi$ across $\mathscr{E}$.*

*2. If there is $g^\star : supp(\Phi) \to [0, 1]$ such that $\Phi$ elicits an $\alpha$-approximately invariant predictor $g^\star \circ \Phi$ across $\mathscr{E}$, then $g^\star \circ \Phi$ is $\sqrt{\alpha/D}$-approximately $\ell_2$ multicalibrated w.r.t. $\mathcal{H}$ and $P_S$, where $D = \min_{e \in \mathscr{E}} P_S(e)$.*

**Remark 3.5.** *(1) In the first statement, assuming $g^\star$ is a bijection avoids degenerate cases where $\Phi$ contains redundant information. For example, every predictor $g^\star \circ \Phi(X)$ upon representation $\Phi$ equals $g^\star(\Phi) \circ \mathbb{I}(X)$ upon representation $X$. Confining $g^\star$ to bijections ensures some unique decomposition into predictors and representations. (2) Wald et al. [41] proves equivalence between exact invariance and simultaneous calibration in each environment. We strengthen their result to show multicalibration on a single source distribution suffices for invariance. Moreover, our results can be directly extended beyond their multi-environment setting to a general uncertainty set of target distributions, by the mapping between grouping functions and density ratios. Further, our theorem is established for both exact and approximate invariance.*

The theorem bridges *approximate multicalibration* with *approximate invariance* for out-of-distribution generalization beyond covariate shift. The equivalence property indicates that the density ratios of target and source distributions constitute the *minimal grouping function class* required for robust prediction in terms of invariance.

# 4 Structure of Grouping Function Classes

Section 3 inspires one to construct richer grouping function classes for stronger generalizability. However, fewer predictors are multicalibrated to a rich function class, and a multicalibrated predictor may not exist at all, as illustrated by the example in section 2.1. In this section, we first study the *maximal grouping function class* that is feasible for a multicalibrated predictor. Then, we will leverage our structural results to inform the design of grouping functions.

## 4.1 Maximal Grouping Function Space

We focus on continuous grouping functions defined on a compact set $\mathcal{X} \times \mathcal{Y} \subset \mathbb{R}^{d+1}$, i.e., $h \in C(\mathcal{X} \times \mathcal{Y})$, and consider absolutely continuous probability measures supported on $\mathcal{X} \times \mathcal{Y}$ with continuous density functions. Our first proposition shows that the maximal grouping function class for any predictor is a linear space.

**Proposition 4.1** (Maximal Grouping Function Class). *Given an absolutely continuous probability measure $P_S(X, Y)$ and a predictor $f : \mathcal{X} \to [0, 1]$, define the maximal grouping function class that $f$ is multicalibrated with respect to:*

$$\mathcal{H}_f := \{h \in C(\mathcal{X} \times \mathcal{Y}) : K_2(f, h, P_S) = 0\}. \tag{7}$$

*Then $\mathcal{H}_f$ is a linear space.*

In the following, we further analyze the spanning set of maximal grouping function classes for nontrivial predictors which are at least calibrated.

**Theorem 4.2** (Spanning Set). *Consider an absolutely continuous probability measure $P_S(X, Y)$ and a calibrated predictor $f : \mathcal{X} \to [0, 1]$. Then its maximal grouping function class $\mathcal{H}_f$ is given by:*

$$\mathcal{H}_f = \mathrm{span}\left\{\frac{p(x, y)}{p_S(x, y)} : p \text{ is continuous and } R_P(f) = \inf_{g:[0,1]\to[0,1]} R_P(g \circ f)\right\}. \tag{8}$$

A predictor's maximal grouping function class is spanned by density ratios of target distributions where the predictor is invariant. Correspondingly, Theorem 3.1 gives the minimal grouping function class, comprised of density ratios between target and source distributions, in order to ensure $f(x)$ is an invariant predictor. In contrast, Theorem 4.2 states the maximal grouping function class for $f(x)$ is exactly the linear space spanned by those density ratios. Next, we further investigate sub-structures of the maximal grouping function class. We focus on the representation learning setting of IRM.

**Theorem 4.3** (Decomposition of Grouping Function Space). *Consider an absolutely continuous probability measure $P_S(X, Y)$ and a measurable function $\Phi : \mathbb{R}^d \to \mathbb{R}^{d_\Phi}$ with $d_\Phi \in Z^+$. We define the Bayes optimal predictor over $\Phi$ as $f_\Phi(x) = \mathbb{E}_{P_S}[Y|\Phi(x)]$. We abbreviate $\mathcal{H}_{f_\Phi}$ with $\mathcal{H}_\Phi$. Then $\mathcal{H}_\Phi$ can be decomposed as a Minkowski sum of $\mathcal{H}_{1,\Phi} + \mathcal{H}_{2,\Phi}$.*

$$\mathcal{H}_{1,\Phi} = \mathrm{span}\left\{\frac{p(\Phi, y)}{p_S(\Phi, y)} : p \text{ is continuous and } R_P(f_\Phi) = \inf_{g:[0,1]\to[0,1]} R_P(g \circ f_\Phi)\right\}. \tag{9}$$

$$\mathcal{H}_{2,\Phi} = \mathrm{span}\left\{\frac{p(x|\Phi, y)}{p_S(x|\Phi, y)} : p \text{ is continuous}\right\}. \tag{10}$$

*1. If a predictor $f$ is multicalibrated with $\mathcal{H}_{1,\Phi}$, then $R_{P_S}(f) \le R_{P_S}(f_\Phi)$.*

*2. $f_\Phi$ is an invariant predictor elicited by $\Phi$ across a set of environments $\mathscr{E}$ where $P_e(\Phi, Y) = P_S(\Phi, Y)$ for any $e \in \mathscr{E}$. If a predictor $f$ is multicalibrated with $\mathcal{H}_{2,\Phi}$, then $f$ is also an invariant predictor across $\mathscr{E}$ elicited by some representation.*

**Remark 4.4.** *$\mathcal{H}_{1,\Phi}$ and $\mathcal{H}_{2,\Phi}$ contain functions defined on $x, \Phi(x), y$ which can both be rewritten as functions on $x, y$ by variable substitution. Thus, $\mathcal{H}_{1,\Phi}, \mathcal{H}_{2,\Phi}$ are still subspaces of grouping functions. $\mathcal{H}_{1,\Phi}$ is spanned by the density ratio of $P(\Phi, Y)$ where the Bayes optimal predictor over $\Phi$ must be invariant on the distribution of $P$. $\mathcal{H}_{2,\Phi}$ is spanned by general density ratio of $P(X|\Phi, Y)$.*

Multicalibration w.r.t. $\mathcal{H}_{1,\Phi}$ ensures at least the *accuracy* of the Bayes optimal predictor on $\Phi$, and multicalibration w.r.t. $\mathcal{H}_{2,\Phi}$ ensures at least the *invariance* of this predictor. However, we show in the following proposition that sizes of two subspaces are negatively correlated. When $\Phi$ is a variable selector, $\mathcal{H}_{1,\Phi}$ expands with more selected covariates while $\mathcal{H}_{2,\Phi}$ shrinks. By choosing a combination of $\mathcal{H}_{1,\Phi}$ and $\mathcal{H}_{2,\Phi}$, we strike a balance between accuracy and invariance of the multicalibrated predictor.

**Proposition 4.5** (Monotonicity). *Consider $X \in \mathbb{R}^d$ which could be sliced as $X = (\Phi, \Psi)^T$ and $\Phi = (\Lambda, \Omega)^T$. Define $\mathcal{H}'_{1,\Phi} := \{h(\Phi(x)) \in C(\mathcal{X} \times \mathcal{Y})\}$, with $\mathcal{H}'_{1,\Phi} \subset \mathcal{H}_{1,\Phi}$. $\mathcal{H}'_{1,X}$ and $\mathcal{H}'_{1,\Lambda}$ are similarly defined. We have:*

*1. $\mathcal{H}'_{1,X} \supset \mathcal{H}'_{1,\Phi} \supset \mathcal{H}'_{1,\Lambda} \supset \mathcal{H}'_{1,\emptyset} = \{C\}$.*

*2. $\{C\} = \mathcal{H}_{2,X} \subset H_{2,\Phi} \subset \mathcal{H}_{2,\Lambda} \subset \mathcal{H}_{2,\emptyset}$.*

*C is a constant value function.*

## 4.2 Design of Grouping Function Classes

The objective of a robust learning method can be represented by a tuple consisting of an assumption about the boundary of distribution shift and a metric of robustness. Multicalibration is equivalent to invariance as a metric of robustness, while the grouping function class provides a unifying view for assumptions over potential distribution shift. Given any uncertainty set of target distributions $\mathcal{P}$, Theorem 4.2 implies an efficient and reasonable construction of grouping functions as linear combinations of density ratios from $\mathcal{P}$. We implement two designs of grouping functions for the learning setting with and without environment annotations respectively.

**From Environments** If samples are drawn from multiple environments and the environment annotations are available, we assume the uncertainty set as the union of each environment's distribution $P_e$. This completely recovers IRM's objective, but we approach it with a different optimization technique in the next section. Taking pooled data as the source $S$, density ratios spanning the grouping function class are $h_e(x,y) = p_e(x,y)/p_S(x,y) = p_S(e|x,y)/p_S(e)$, where $p_S(e|x,y)$ is estimated by an environment classifier. Then a grouping function can be represented as a linear combination of $h_e$:

$$h(x,y) = \sum_{e \in \mathscr{E}} \lambda_e p_S(e|x,y), \quad \lambda_e \in \mathbb{R}. \tag{11}$$

**From Hard Samples** When data contains latent sub-populations without annotations, the uncertainty set can be constructed by identifying sub-populations. Hard sample learning [27, 28, 29] suggests the risk is an indicator for sub-population structures. Samples from the minority sub-population $M$ are more likely to have high risks. For example, JTT [29] identified the minority subgroup using a risk threshold of a trained predictor $f_{id}$. We adopt a continuous grouping by assuming $P_S(X, Y \in M) \propto (f_{id}(X) - Y)^2$. We construct the uncertainty set as the union of the source $S$ and minority sub-population $M$, resulting in a grouping function represented as:

$$h(x,y) = \lambda_M (f_{id}(x) - y)^2 + \lambda_S, \quad \lambda_M, \lambda_S \in \mathbb{R}. \tag{12}$$

Another design utilizing Distributionally Robust Optimization's assumption [10] is in section B.

## 5 MC-PseudoLabel: An Algorithm for Extended Multicalibration

In this section, we introduce an algorithm for multicalibration with respect to joint grouping functions. Simultaneously, the algorithm also provides a new optimization paradigm for invariant prediction under distribution shift. The algorithm, called MC-PseudoLabel, post-processes a trained model by supervised learning with pseudolabels generated by grouping functions. As shown in Algorithm 1, given a predictor function class $\mathcal{F}$ and a dataset $D$ with an empirical distribution $\hat{P}_D(X, Y)$, a regression oracle $A$ solves the optimization: $A_{\mathcal{F}}(D) = \arg\min_{f \in \mathcal{F}} R_{\hat{P}_D}(f)$. We take as input a model $f_0$, possibly trained by Empirical Risk Minimization. $f_0$ has a finite range following conventions of prior work in multicalibration [13]. For continuous predictors, we discretize the model output and introduce a small rounding error (see section C). For each iteration, the algorithm performs regression with grouping functions on each level set of the model. The prediction of grouping functions rectify the uncalibrated model and serves as pseudolabels for model updates.

Since we regress $Y$ with grouping functions defined on $Y$, a poor design of groupings violating Theorem 4.2 can produce trivial outputs. For example, if grouping functions contain $h(x,y) = y$, then $err_{t-1} - \tilde{err}_t$ never decreases and the algorithm outputs $f_0$, because there does not exist a multicalibrated predictor. However, the algorithm certifies a multicalibrated output if it converges.

**Theorem 5.1** (Certified Multicalibration). *In Algorithm 1, for $\alpha, B > 0$, if $err_{t-1} - \tilde{err}_t \leq \frac{\alpha}{B}$, the output $f'_{t-1}(x)$ is $\alpha$-approximately $\ell_2$ multicalibrated w.r.t. $\mathcal{H}_B = \{h \in \mathcal{H} : \sup h(x,y)^2 \leq B\}$.*

---

**Algorithm 1** MC-PseudoLabel

---

**Require:** A dataset $D = (D_x, D_y)$, a grouping function class $\mathcal{H}$, a predictive function class $\mathcal{F}$.

1:   $t \leftarrow 0$;
2:   $f_0 \leftarrow$ Initialization;     {*For example, models trained with ERM.*}
3:   $m \leftarrow |\text{Range}(\text{Discretize}(f_0))|$;
4:   **repeat**
5:     $f_t' \leftarrow \text{Round}(f_t; m) := \arg\min_{v \in [1/m]} |f_t(x) - v|$;
6:     $err_t = \mathbb{E}_{x,y \sim D}[(f_t'(x) - y)^2]$;
7:     **for** each $v \in [1/m]$ **do**
8:       $D_v^t \leftarrow D | f_t'(x) = v$;
9:       $h_v^t(x, y) \leftarrow A_{\mathcal{H}}(D_v^t)$;     {*Regression on level sets with grouping functions.*}
10:    **end for**
11:    $\tilde{f}_{t+1}(x, y) \leftarrow \sum_{v \in [\frac{1}{m}]} 1_{\{f_t'(x) = v\}} \cdot h_v^t(x, y)$;     {*Generate pseudolabels.*}
12:    $\tilde{err}_{t+1} \leftarrow \mathbb{E}_{x,y \sim D}[(\tilde{f}_{t+1}(x, y) - y)^2]$;
13:    $D_{t+1} \leftarrow (D_x, \tilde{f}_{t+1}(D))$;
14:    $f_{t+1}(x) \leftarrow A_{\mathcal{F}}(D_{t+1})$;     {*Update the model with pseudolabels.*}
15:    $t \leftarrow t + 1$;
16: **until** $err_{t-1} - \tilde{err}_t$ stops decreasing.

**Ensure:** $f_{t-1}'(x)$.

---

MC-PseudoLabel reduces to LSBoost Globus-Harris et al. [13], a boosting algorithm for multicalibration if $\mathcal{H}$ only contains covariate-based grouping functions. In this case, Line 14 of Algorithm 1 reduces to $f_{t+1}(x) = \tilde{f}_{t+1}(x, \cdot)$ where $\tilde{f}_{t+1}$ does not depend on $y$. For joint grouping functions, since $\tilde{f}_{t+1} \in \mathbb{R}^{\mathcal{X} \times \mathcal{Y}}$, we project it to models' space of $\mathbb{R}^{\mathcal{X}}$ by learning the model with $\tilde{f}_{t+1}$ as pseudolabels. The projection substantially changes the optimization dynamics. LSBoost constantly decreases risks of models, due to $R_{\hat{P}_D}(f_{t+1}) = R_{\hat{P}_D}(\tilde{f}_{t+1}) < R_{\hat{P}_D}(f_t)$. The projection step disrupts the monotonicity of risks, implying that MC-Pseudolabel can output a predictor with a higher risk than input. This is because multicalibration with joint grouping functions implies balance between accuracy and invariance, as is discussed in Theorem 4.3. The convergence of LSBoost relies on the monotonicity of risks, which is not applicable to MC-Pseudolabel. We study the algorithm's convergence in the context of representation learning. Assume we are given a grouping function class $\mathcal{H}_\Phi$ with a latent representation $\Phi$. If a predictor is multicalibrated w.r.t $\mathcal{H}_{1,\Phi}, \mathcal{H}_{2,\Phi}$ respectively, then it is also multicalibrated w.r.t. $\mathcal{H}_\Phi$. Therefore, we separately study the convergence with two grouping function classes. In Proposition F.26, we show the convergence for a subset of $\mathcal{H}_{1,\Phi}$ consisting of covariate-based grouping functions, which is a corollary of Globus-Harris et al.'s result. As a greater challenge, we derive convergence for $\mathcal{H}_{2,\Phi}$ when data follows multivariate normal distributions.

**Theorem 5.2** (Covergence for $\mathcal{H}_{2,\Phi}$ (informal, see Theorem F.27)). *Consider $X \in \mathbb{R}^d$ with $X = (\Phi, \Psi)^T$. Assume that $(\Phi, \Psi, Y)$ follows a multivariate normal distribution $\mathcal{N}_{d+1}(\mu, \Sigma)$ where the random variables are in general position such that $\Sigma$ is positive definite. For any distribution $D$ supported on $\mathcal{X} \times \mathcal{Y}$, take the predictor class $\mathcal{F} = \mathbb{R}^{\mathcal{X}}$ and the grouping function class $\mathcal{H}$ as a subset of $\mathcal{H}_{2,\Phi}$ which is defined in Equation 10:*

$$\mathcal{H} = \{h : h \in \mathcal{H}_{2,\Phi} \text{ and } h(x, y) = c_x^T x + c_y y + c_b, c_x \in \mathbb{R}^d, c_y, c_b \in \mathbb{R}\}. \tag{13}$$

*For an initial predictor $f_0(x) = \mathbb{E}[Y|x]$, run MC-Pseudolabel$(D, \mathcal{H}, \mathcal{F})$ without rounding, then $f_t(x)$ converges pointwise to $\mathbb{E}[Y|\Phi(x)]$ as $t \to \infty$, with a convergence rate of $\mathcal{O}(M(\Sigma)^t)$ where $0 \leq M(\Sigma) < 1$.*

MC-Pseudolabel is also an optimization paradigm for invariance. Certified multicalibration in Theorem 5.1 also implies certified invariance. Furthermore, MC-Pseudolabel introduces no extra hyperparameters to tradeoff between risks and robustness. Both certified invariance and light-weighted hyperparameters simplify model selection, which is challenging for out-distribution generalization because of unavailable validation data from target distributions [15]. MC-Pseudolabel has light-weighted optimization consisting of a series of supervised regression. It introduces an overhead to Empirical Risk Minimization by performing regression on level sets. However, the extra burden is linear regression by designing the grouping function class as linear space. Furthermore, regression on different level sets can be parallelized. Computational complexity is further analyzed in section D.

# 6 Experiments

## 6.1 Settings

We benchmark MC-Pseudolabel on real-world *regression* datasets with distributional shift. We adopt two experimental settings. For the *multi-environment* setting, algorithms are provided with training data collected from multiple annotated environments. Thereafter, the trained model is assessed on new environments. For the *single-environment* setting, algorithms are trained on a single source distribution. There could be latent sub-populations in training data, but environment annotations are unavailable. The trained model is assessed on a target dataset with distribution shift from the source. The grouping function class is implemented according to Equation 11 and Equation 12 for the multi-environment and single-environment setting respectively.

**Datasets**   We experiment on PovertyMap [44] and ACSIncome [7] for the multi-environment setting, and VesselPower [33] for the single-environment setting. As the only regression task in WILDS [23], a popular benchmark for in-the-wild distribution shift, PovertyMap performs poverty index estimation for different spatial regions by satellite images. Data are collected from both urban and rural regions, by which the environment is annotated. The test dataset also covers both environments, but is collected from different countries. The primary metric is *Worst-U/R Pearson*, the worst Pearson correlation of prediction between rural and urban regions. The other two datasets are tabular, where natural concept shift ($Y|X$ shift) is more common due to existence of missing variables and hidden confounders [31]. ACSIncome [7] performs personal income prediction with data collected from US Census sources across different US states. The task is converted to binary classification by an income threshold, but we take raw data for regression. Environments are partitioned by different occupations with similar average income. VesselPower comes from Shifts [32, 33], a benchmark focusing on regression tasks with real-world distributional shift. The objective is to predict power consumption of a merchant vessel given navigation and weather data. Data are sampled under different time and wind speeds, causing distribution shift between training and test data.

**Baselines**   For the multi-environment setting, baselines include ERM (Empirical Risk Minimization); methods for invariance learning which mostly adopts multi-objective optimization: IRM [2], MIP [24], IB-IRM [1], CLOvE [41], MRI [18], REX [25], Fishr [36]; an alignment-based method from domain generalization: IDGM [39]; and Group DRO [38]. Notably, CLOvE learns a calibrated predictor simultaneously on all environments, but it is optimized by multi-objective learning with a differentiable regularizer for calibration. For the singe-environment setting, baselines include reweighting based techniques: CVaR [26], JTT [29], Tilted-ERM [27]; a Distributionally Robust Optimization method $\chi^2$-DRO [8]; and a data augmentation method C-Mixup [43]. Other methods are not included because of specification in classification [45, 46] or exposure to target distribution data during training [19, 22]. For all experiments, we train an Oracle ERM with data sampled from target distribution.

**Implementation**   We implement the predictor with MLP for ACSIncome and VesselPower, and Resnet18-MS [16] for PovertyMap, following WILDS' default architecture. We follow DomainBed's protocol [15] for *model selection*. Specifically, we randomly sample 20 sets of hyperparameters for each method, containing both the training hyperparameters and extra hyperparameters from the robust learning algorithm. We select the best model across hyperparameters based on three model selection criteria, including in-distribution validation on the average of training data, worst-environment validation with the worst performance across training environments, and oracle validation on target data. Oracle validation is not recommended by DomainBed, which suggests limited numbers of access to target data. The entire run is repeated with different seeds for three times to measure standard errors of performances. Specifically for PovertyMap, we perform 5-fold cross validation instead of three repeated experiments, following WILDS' setup.

## 6.2 Results

Results are shown in Table 1 for multi-environment settings and Table 2 for single-environment settings. MC-Pseudolabel achieves superior performance in all datasets with in-distribution and worst-environment validation which does not violate test data. For oracle validation, MC-Pseudolabel achieves comparable performances to the best method. For example, CLOvE, which also learns invariance by calibration, achieves best performance under oracle validation in PovertyMap, but it sharply degrades when target validation data is unavailable. It's

Table 1: Results on multi-environment datasets, evaluated on test data using three model selection criteria. ID: validation with averaged performance on training data. Worst: validation with the worst performance across training environments. Oracle: validation with performance on sampled test set.

| Method | ACSIncome: RMSE ↓ | | | PovertyMap: Worst-U/R Pearson ↑ | | |
|---|---|---|---|---|---|---|
| | ID | Worst | Oracle | ID | Worst | Oracle |
| ERM | $0.487_{\pm 0.009}$ | $0.487_{\pm 0.009}$ | $0.452_{\pm 0.012}$ | $0.48_{\pm 0.06}$ | $0.48_{\pm 0.06}$ | $0.49_{\pm 0.07}$ |
| IRM | $0.466_{\pm 0.002}$ | $0.466_{\pm 0.002}$ | $0.465_{\pm 0.002}$ | $0.38_{\pm 0.07}$ | $0.39_{\pm 0.06}$ | $0.45_{\pm 0.08}$ |
| MIP | $0.457_{\pm 0.008}$ | $0.454_{\pm 0.012}$ | $0.454_{\pm 0.012}$ | $0.40_{\pm 0.09}$ | $0.39_{\pm 0.10}$ | $0.43_{\pm 0.08}$ |
| IB-IRM | $0.463_{\pm 0.003}$ | $0.463_{\pm 0.003}$ | $0.438_{\pm 0.009}$ | $0.39_{\pm 0.07}$ | $0.37_{\pm 0.05}$ | $0.43_{\pm 0.06}$ |
| CLOvE | $0.455_{\pm 0.005}$ | $0.454_{\pm 0.002}$ | $0.450_{\pm 0.005}$ | $0.46_{\pm 0.09}$ | $0.42_{\pm 0.13}$ | $\mathbf{0.51}_{\pm 0.06}$ |
| MRI | $0.458_{\pm 0.011}$ | $0.458_{\pm 0.011}$ | $0.455_{\pm 0.013}$ | $0.47_{\pm 0.10}$ | $0.46_{\pm 0.08}$ | $0.49_{\pm 0.07}$ |
| REX | $0.466_{\pm 0.009}$ | $0.464_{\pm 0.009}$ | $0.458_{\pm 0.003}$ | $0.43_{\pm 0.09}$ | $0.42_{\pm 0.09}$ | $0.45_{\pm 0.09}$ |
| Fishr | $0.458_{\pm 0.006}$ | $0.455_{\pm 0.012}$ | $0.450_{\pm 0.008}$ | $0.42_{\pm 0.09}$ | $0.41_{\pm 0.09}$ | $0.43_{\pm 0.08}$ |
| IDGM | $1.843_{\pm 0.018}$ | $1.843_{\pm 0.018}$ | $1.843_{\pm 0.018}$ | $0.02_{\pm 0.07}$ | $0.01_{\pm 0.15}$ | $0.13_{\pm 0.14}$ |
| GroupDRO | $0.481_{\pm 0.035}$ | $0.449_{\pm 0.017}$ | $0.433_{\pm 0.013}$ | $0.38_{\pm 0.15}$ | $0.37_{\pm 0.16}$ | $0.42_{\pm 0.12}$ |
| MC-Pseudolabel | $\mathbf{0.428}_{\pm 0.009}$ | $\mathbf{0.425}_{\pm 0.012}$ | $\mathbf{0.411}_{\pm 0.011}$ | $\mathbf{0.50}_{\pm 0.06}$ | $\mathbf{0.50}_{\pm 0.06}$ | $0.50_{\pm 0.06}$ |
| Oracle ERM | | | $0.332_{\pm 0.001}$ | | | $0.71_{\pm 0.05}$ |

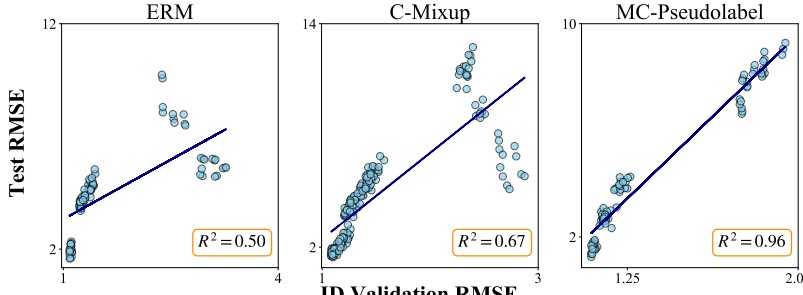

Figure 1: Accuracy-on-the-line beyond covariate shift: correlation between models' in-distribution and out-of-distribution risks on VesselPower.

because CLOvE tunes its regularizer's coefficient to tradeoff with ERM risk, whose optimal value depends on the target distribution shift. In contrast, MC-Pseudolabel exhibits an advantage with in-distribution model selection. This is further supported by Figure 6.2, which shows that MC-Pseudolabel's out-of-distribution errors strongly correlates with in-distribution errors. The experiment spans across different hyperparameters and seeds with the same model architecture on VesselPower. The phenomenon, known as accuracy-on-the-line [34], is well known for a general class of models under covariate shift. However, Liu et al. [31] shows accuracy-on-the-line does not exist under concept shift, which is the case for ERM and C-Mixup. This introduces significant challenge for model selection. However, MC-Pseudolabel recovers the accuracy to the line.

Table 2: Single-environment results.

| Method | VesselPower: RMSE ↓ | |
|---|---|---|
| | ID | Oracle |
| ERM | $1.92_{\pm 0.23}$ | $1.86_{\pm 0.19}$ |
| CVaR | $1.69_{\pm 0.18}$ | $\mathbf{1.49}_{\pm 0.10}$ |
| JTT | $1.75_{\pm 0.27}$ | $1.58_{\pm 0.15}$ |
| Tilted-ERM | $1.72_{\pm 0.21}$ | $1.61_{\pm 0.12}$ |
| $\chi^2$-DRO | $1.69_{\pm 0.20}$ | $1.56_{\pm 0.06}$ |
| C-Mixup | $1.72_{\pm 0.15}$ | $1.56_{\pm 0.08}$ |
| MC-Pseudolabel | $\mathbf{1.61}_{\pm 0.20}$ | $1.52_{\pm 0.16}$ |
| Oracle ERM | | $1.18_{\pm 0.01}$ |

# 7 Conclusion

To conclude, we establish a new optimization framework for out-of-distribution generalization through extended multicalibration with joint grouping functions. While the current algorithm focuses on regression, there is potential for future work to extend our approach to general forms of tasks, particularly in terms of classification.

## Acknowledgments and Disclosure of Funding

Peng Cui is supported in part by National Natural Science Foundation of China (No. 62425206, 62141607).

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

# A  Related Work

**Multicalibration**  Multicalibration is first proposed by Hébert-Johnson et al. [17] with binary grouping functions. Kim et al. [20] and Gopalan et al. [14] extend the grouping functions to real-valued functions. Globus-Harris et al. [13] shows that with a sufficiently rich class of real-valued grouping functions, multicalibration actually implies accuracy. Globus-Harris et al. also provides a boosting algorithm for both regression and multicalibration. The connection between multicalibration and distribution shift is first studied by Kim et al. [21], who proves that $\ell_1$ multicalibration error remains under covariate shift, given a sufficiently large real-valued grouping function class. Kim et al. further shows that under covariate shift, a multicalibrated predictor can perform statistical inference of the average outcome of a sample batch. In contrast, we derive a robustness result for individual prediction of outcomes for $\ell_2$ multicalibrated predictors. In addition, Wald et al. [41] studies the equivalence of Invariant Risk Minimization and simultaneous calibration on each environment. Our equivalence results for multicalibration can be perceived as a generalization of Wald et al.'s results beyond the multi-environment setting, by deriving a mapping between density ratios and grouping functions. We also extend the equivalence to approximately multicalibrated and approximately invariant predictors. Furthermore, we move beyond Wald et al.'s multi-objective optimization with Lagrangian regularization, by proposing a new post-processing optimization framework consisting of a series of supervised regression. Meanwhile, Blasiok et al. [4] discusses connections between calibration and post-processing, which is an equivalent expression of invariance. There are other extensions of multicalibration, such as Deng et al. [6] who generalize the term $Y - f(X)$ in multicalibration's definition to a class of general functions. While our work is the first to generalize the grouping functions $h$ to consider the outcomes.

**Out-of-distribution Generalization Beyond Covariate Shift**  Despite abundant literature from domain generalization that focuses on image classification where covariate shift dominates, research on algorithmic robustness on regression tasks beyond covariate shift is relatively limited. The setting can be categorized according to if the source distribution is partitioned into several environments. For the multi-environment generalization setting, Invariant Risk Minimization and its variants assume that outcomes are generated by a common causal structural equation across all environments, and aims to recover such an invariant (or causal) predictor [1, 2, 18, 24, 25, 36]. Group DRO [38] is a simple but surprisingly strong technique that optimizes for the worst group risk with reweighting of environments. There are also meta-learning methods [12] that handles multi-environment generalization with bi-level optimization. For the single environment setting, Distributionally Robust Optimization optimizes for the worst-case risk in an uncertainty set of distributions centering around the source distribution [8, 9, 11, 26, 40]. Another branch of research is targeted at mitigating spurious correlation with an assumption of simplicity bias, which utilizes a simple model to discover latent sub-populations and then correct the biased predictor by sample reweighting [27, 28, 29], retraining on a subgroup-balanced dataset or a small batch from target distribution [19, 22, 46], or perform Invariant Risk Minimization on discovered subgroups [5]. Data augmentation is a prevalent technique to enhance algorithmic robustness for vision tasks. Quite a lot of these methods are tailored for classification. For example, Mixup [45] interpolates between features of samples with the same label. The approach is extended to regression settings by C-Mixup [43]. Pseudolabelling is a common technique for out-of-distribution generalization, but typically adopted in a setting with exposure to unlabelled samples from target distribution, known as domain adaptation [42]. However, MC-Pseudolabel generate pseudolabels for the source distribution itself.

# B  Grouping Functions for Distributionally Robust Optimization

Distributionally Robust Optimization assumes the target distribution to reside in an uncertainty set $\mathcal{P}$ of distributions centering around the source distribution $P_S$. For example, Duchi et al. [10] formulates the uncertainty set as arbitrary subgroups that has a proportion of at least $\alpha_0 \in (0, 1)$. Duchi et al. only consider subgroups of covariates:

$$\mathcal{P}(X) = \{P(X) : \text{there exists a probability measure } P'(X), \tag{14}$$

$$P_S(X) = \alpha P(X) + (1 - \alpha)P'(X), \alpha \geq \alpha_0\}. \tag{15}$$

By the correspondence between density ratios and grouping functions, the equivalent design of a grouping function class is given by:

$$\mathcal{H} = \left\{ h \in \mathbb{R}^{\mathcal{X}} : 0 \leq h(x) \leq \frac{1}{\alpha_0}, \ \ \forall x \right\}. \tag{16}$$

We can also extend the grouping functions to consider both covariates and outcomes, such that general subgroups are incorporated into the uncertainty set:

$$\mathcal{H} = \left\{ h \in \mathbb{R}^{\mathcal{X} \times \mathcal{Y}} : 0 \leq h(x, y) \leq \frac{1}{\alpha_0}, \ \ \forall x, y \right\}. \tag{17}$$

In the case of grouping functions defined on $x$ and $y$ jointly, the grouping function class is not closed under affine transformation and is not a linear space spanned by density ratios, which suggests that a perfectly multicalibrated solution might not exist. However, approximately multicalibrated predictors can still be pursued.

## C   Model Discretization

For continuous predictors, we take a preprocessing step to discretize the model to as many bins as possible such that the rounding error is negligible while still ensuring enough samples in individual bins. Specifically, we equally split the outcomes of predictors to bins with equal intervals from the minimum to maximum of model output. We start from a minimum bin number $m = 10$, and keeps increasing $m$ as long as $90\%$ of the samples reside in a bin with at least 30 samples. When the criterion is violated, we stop increasing $m$ and select it as the final bin number. The model discretization procedure is fixed across all experiments.

## D   Computational Complexity

We assume that the predictor's outcomes are uniformly distributed. Denote the average bin size by $N_b$, which is a constant around 30 in our implementation. The bin number is given by $m = N/N_b$ where $N$ is the sample size. For neural networks, $N$ represents the batch size. The overhead of MC-Pseudolabel compared to Empirical Risk Minimization is linear regression on each bin, whose sample complexity is $\mathcal{O}(N_b^3)$ with OLS. Please note that an individual linear regression for around 30 samples is extremely cheap. A non-parallel implementation of regression on every bin scales linearly with the bin number $m$, so the overall complexity is $\mathcal{O}(N_b^2 N)$. However, since the regression on each bin is independent, we adopt a multi-processing implementation. Denote the number of jobs by $J$, the overall time cost of MC-Pseudolabel is $\mathcal{O}(N_b^2 N/J)$. As a comparison, OLS on $N$ samples has a computational complexity of $\mathcal{O}(N^3)$.

In conclusion, the complexity of MC-Pseudolabel scales linearly with sample size (or batch size for neural networks). Counterintuitively, increasing the bin number $m$ (and thus decreasing the bin size) actually decreases the computational complexity. This is because linear regression scales cubically with sample size, so decreasing the sample size in each bin is preferred to decreasing the bin number.

## E   Experiments

### E.1   An Additional Experiment: Synthetic Dataset

We start from a multi-environment synthetic dataset with a multivariate normal distribution corresponding to Theorem 5.2. In this experiment, we examine the optimization dynamics of MC-Pseudolabel. The data generation process is inspired by Arjovsky et al. [2]. The covariates can be sliced into $X = (S, V)^T$ with $S \in \mathbb{R}^9$ and $V \in \mathbb{R}$, where $S$ is the causal variable for $Y$ and $V$ is the spurious variable. The data is generated by the following structural equations:

$$S \sim \mathcal{N}(0, 1). \tag{18}$$

$$Y = \alpha_S^T S + \epsilon_Y, \quad \alpha_S = (1, ..., 1)^T \in \mathbb{R}^9, \epsilon_Y \sim \mathcal{N}(0, 0.5^2). \tag{19}$$

$$V = \alpha_V(\mathcal{E}) \cdot Y + \epsilon_V, \quad \alpha_V(e_1) = 1.25, \alpha_V(e_2) = 0.75, \alpha_V(e_T) = -1, \epsilon_V \sim \mathcal{N}(0, 0.1^2). \tag{20}$$

The covariate $V$ is spuriously correlated with $Y$ because the coefficient $\alpha_V(e)$ depends on the specific environment $e$. We set $\alpha_V(\mathcal{E}) = 1.25, 0.75$ respectively for two training environments while $\alpha_V(\mathcal{E})$ extrapolates to $-1$ during testing. A robust algorithm is supposed to bypass the spurious variable and output a predictor $f(X) = \alpha_S^T S$ in order to survive the test distribution where the correlation between $V$ and $Y$ is negated.

The predictor class for this dataset is linear models, and the environment classifier is implemented by MLP with a single hidden layer. In this experiment, we fix the training hyperparameters for the base linear model, and perform grid search over the extra hyperparameter introduced by robust learning methods. Baselines except for ERM and Group DRO share a hyperparameter which is the regularizer's coefficient, and Group DRO introduces a temperature hyperparameter. We search over their hyperparameter space and report RMSE metric on the test set in Figure 2. Most baselines exhibit a U-turn with an increasing hyperparameter, and the minimum point varies across methods. The sensitivity of hyperparameters implies the dependence on a strong model selection criterion, such as oracle model selection on target distribution. However, the dashed line for MC-Pseudolabel's error is tangent to all the U-turns of baselines, indicating a competitive performance of MC-Pseudolabel both with and without oracle model selection.

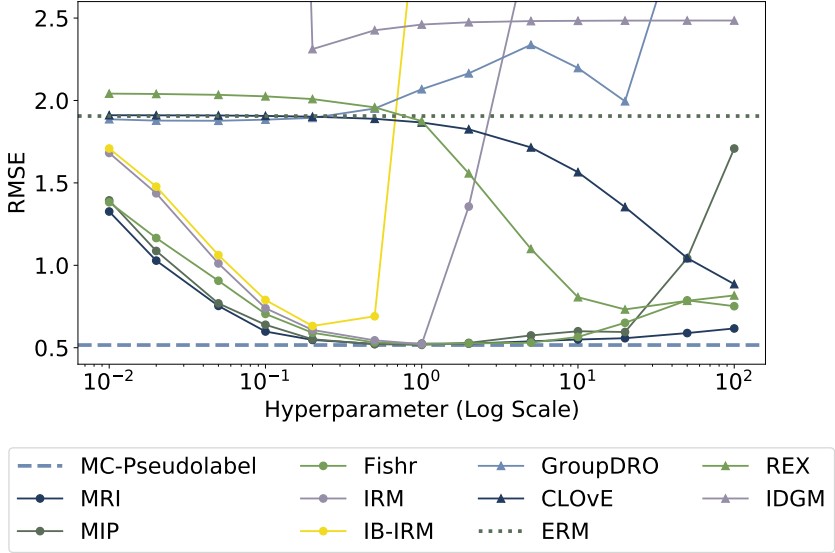

Figure 2: Results (RMSE) on the synthetic dataset. Curves show method performances across hyperparameters. Methods without extra hyperparameters are marked with dotted lines.

We also investigate the evolution of pseudo labels $\tilde{f}_t$ in Algorithm 1 to recover the dynamics of MC-Pseudolabel. The first row of Figure 3 demonstrates how pseudolabelling results in a multicalibrated predictor. It shows that pseudolabels for two environments deviate from model prediction at Step 0, but the gap quickly converges at Step 4, implying multicalibrated prediction. The second row provides insight about how pseudolabelling contributes to an invariant predictor. We observe that the curve of two environments are gradually merging because the pseudolabel introduces a special noise to the original label such that the correlation between the pseudolabel and spurious variable $V$ is weakened. As a result, the predictor will depend on the causal variable $S$ which is relatively more strongly correlated with pseudolabels.

## E.2  VesselPower

In figure 4, we provide the correlation between models' in-distribution validation performance and out-of-distribution test performance across all methods.

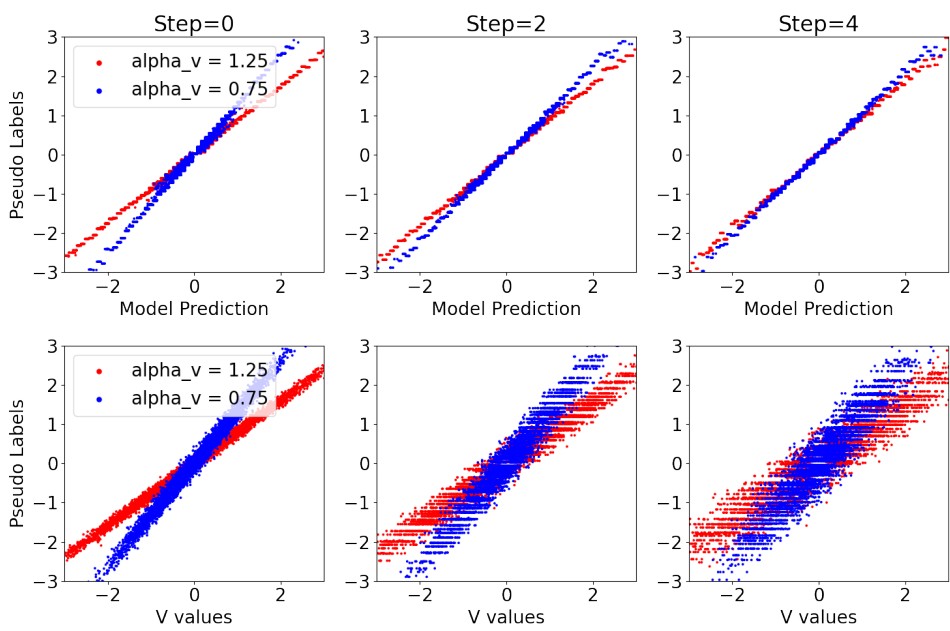

Figure 3: Evolution of pseudolabels during MC-Pseudolabel. The first row plots values of pseudolabels against model predictions. The second row plots values of pseudolabels against $V$. Columns represent different snapshots during optimization.

### E.3  Training Details

**Datasets.** In the ACSInome dataset, we focus on predicting personal income for California residents across four occupation fields: science professions (SCI), protective services (PRT), education (EDU), and military (MIL). While average incomes are similar across these fields, the correlation between income and usual hours worked per week (WKHP) varies significantly, as shown in Table 3. This spurious correlation between income and WKHP introduces a concept shift across occupations. Therefore, we train the model on environments comprising the SCI, PRT, and EDU occupations and evaluate it on MIL.

Table 3: Statistics of ACSIncome (California)

|  | SCI | PRT | EDU | MIL |
|---|---|---|---|---|
| Average Income ($10K) | 4.8 | 4.6 | 4.5 | 4.5 |
| Pearsonr (Income, WKHP) | 0.38 | 0.51 | 0.59 | -0.30 |

We follow the standard setup for PovertyMap as specified in the WILDS benchmark [23]. A natural spatial distribution shift occurs between data collected from urban and rural regions, both of which are represented in the training and test sets. However, the test data is sourced from different countries than the training data. The primary evaluation metric is the worst Pearson correlation between predictions in urban and rural environments.

We use the synthetic split of VesselPower where the target power labels are generated through the simulation of a physics model based on read data input features. The dataset include a single environment training set, a dev-in set for in-distribution validation, and a dev-out set for evaluation. Distribution shifts arise due to variation in time and wind speeds. We report RMSE of prediction in megawatts (MW), rather than the kilowatts (KW) as used in the dataset's original paper [33].

**Model Selction.** We follow DomainBed's protocol [15] for model selection. Specifically, we randomly sample 20 sets of hyperparameters for each method, containing both the training hyperparameters of base models in Table 4 and extra hyperparameters from the robust learning algorithm

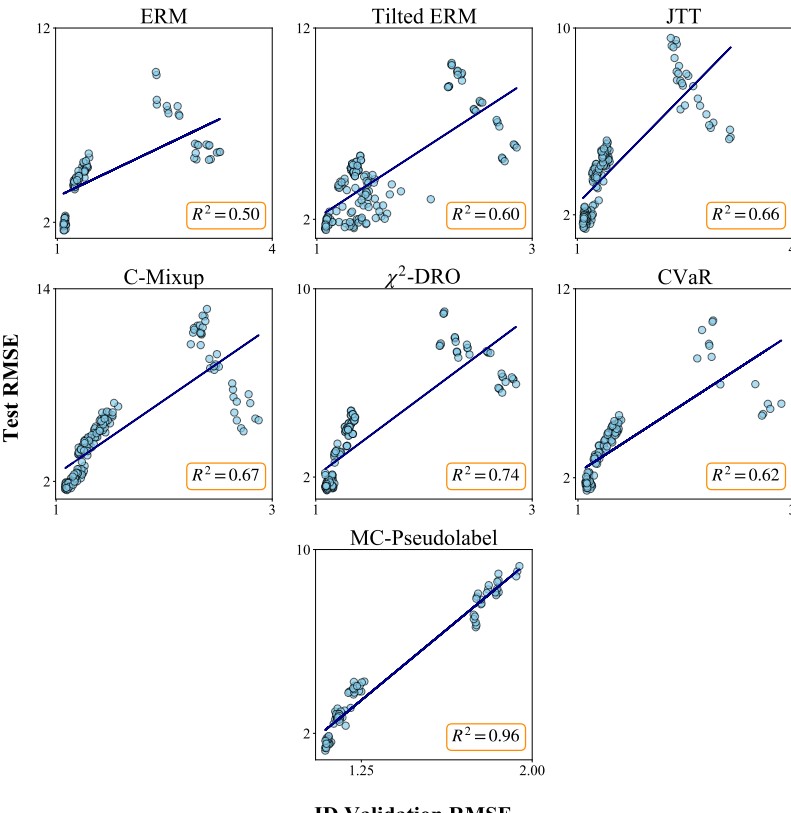

Figure 4: Correlation between models' in-distribution and out-of-distribution risks on VesselPower.

in Table 5. We select the best model across hyperparameters based on three model selection criteria. In-distribution (ID) validation selects the model with the best metric on the average of an in-distribution validation dataset, which is sampled from the same distribution as the training data. Worst-environment (Worst) validation selects the best model by the worst performance across all environments in the in-distribution validation dataset. Worst validation is applicable only to the multi-environment setting. Oracle validation selects the best model by an out-of-distribution validation dataset sampled from the target distribution of test data. Oracle validation leaks the test distribution, so it is not recommended by DomainBed. However, most robust learning methods relies on out-of-distribution validation, so Domainbed suggests limited numbers of access to target data when using Oracle validation. Though MC-Pseudolabel already performs well under ID and Worst validation, we still report its performance under Oracle validation to compare the limit of robust learning methods regardless of model selection.

Following DomainBed, the entire model selection procedure is repeated with different seeds for three times to measure standard errors of performances. Thus, we have totally 60 runs per method per dataset. Specifically for PovertyMap, we follow WILDS' setup [23] and perform 5-fold cross validation instead of three repeated experiments. For each fold of the dataset, we conduct the model selection procedure four times across three seeds, summing up to a total of 12 experiments. And we report the average and standard error of performances across 5 folds. Thus, the standard error measures both the difficulty disparity across folds and the model's instability.

**Grouping Functions.** The grouping function class of MC-Pseudolabel is implemented according to Equation 11 and Equation 12 for the multi-environment and single-environment setting respectively. For the multi-environment setting, the environment classifier $p(e|x, y)$ is implemented as MLP with

a single hidden layer of size 100 for tabular datasets including Simulation and ACSIncome. For PovertyMap, the environment classifier is implemented by Resnet18-MS with the same architecture as the predictor, except that the label $y$ is fed into the last fully-connected layer. For the single-environment setting of VesselPower, the identification model $f_{id}$ is implemented as a Ridge regression model.

Table 4: Hyperparameters for model architecture.

|  | Simulation | ACSIncome | VesselPower | PovertyMap |
|---|---|---|---|---|
| Architecture | Linear | MLP | MLP | Resnet18-MS |
| Hidden Layer Dimensions | None | 16 | 32, 8 | Standard [23] |
| Optimizer | Adam | Adam | Adam | Adam |
| Weight Decay | 0 | 0 | 0 | 0 |
| Loss | MSE | MSE | MSE | MSE |
| Learning Rate | 0.1 | $10^{\text{Uniform}[-3,-1]}$ | $10^{\text{Uniform}[-3,-1]}$ | $0.001^1$ |
| Batch Size | 1024 | [256, 512, 1024, 2048] | [256, 512, 1024, 2048] | $64^1$ |

[1] The learning rate and batch size for training ResNet follow the setup of WILDS [23].

Table 5: Hyperparameters for robust learning methods.

|  | Range |
|---|---|
| Regularizer Coefficient | $10^{\text{Uniform}[-3,2]}$ |
| $\eta$ (GroupDRO) | $10^{\text{Uniform}[-3,2]}$ |
| $\alpha$ (JTT) | [0.1, 0.2, 0.5, 0.7] |
| $\lambda$ (JTT) | [5, 10, 20, 50] |
| $\eta$ ($\chi^2$-DRO) | [0.2, 0.5, 1, 1.5] |
| $t$ (Tilted-ERM) | [0.1, 0.5, 1, 5, 10, 50, 100, 200] |
| $\alpha$ (C-Mixup) | [0.5, 1, 1.5, 2] |
| $\sigma$ (C-Mixup) | [0.01, 0.1, 1, 10, 100] |

### E.4  Software and Hardware

Our experiments are based on the architecture of PyTorch [35]. Each experiment with a single set of hyperparameters is run on one NVIDIA GeForce RTX 3090 with 24GB of memory, taking at most 15 minutes.

## F  Theory

### F.1  Multicalibration and Bayes Optimality under Covariate Shift

**Assumption F.1** (Restatement of Assumption 2.3)**.**

*1. (Closure under Covariate Shift) For a set of probability measures $\mathcal{P}(X)$ containing the source measure $P_S(X)$,*

$$\forall P \in \mathcal{P}, h \in \mathcal{H} \Rightarrow \frac{p(\cdot)}{p_S(\cdot)} \cdot h(\cdot) \in \mathcal{H}. \tag{21}$$

*2. (($\gamma, \rho$)-Weak Learning Condition) For any $P \in \mathcal{P}(X)P_S(Y|X) \equiv \{P'(X)P_S(Y \mid X) : P' \in \mathcal{P}\}$ with the source conditional measure $P_S(Y|X)$ and every measurable set $G \subset \mathcal{X}$ satisfying $P(X \in G) > \rho$, if*

$$\mathbb{E}_P[(\mathbb{E}_P[Y|X] - Y)^2|X \in G] < \mathbb{E}_P[(\mathbb{E}_P[Y|X \in G] - Y)^2|X \in G] - \gamma, \tag{22}$$

*then there exists $h \in \mathcal{H}$ satisfying*

$$\mathbb{E}_P[(h(X) - Y)^2|X \in G] < \mathbb{E}_P[(\mathbb{E}_P[Y|X \in G] - Y)^2|X \in G] - \gamma. \tag{23}$$

**Lemma F.2** (Globus-Harris et al. [13]). *Fix any distribution $P \in \mathcal{P}(X, Y)$, any model $f : \mathcal{X} \to [0, 1]$, and any class of real valued functions $\mathcal{H}$ that is closed under affine transformation. Let:*

$$\mathcal{H}_1 = \{h \in \mathcal{H} : \max_{x \in \mathcal{X}} h(x)^2 \leq 1\}$$

*be the set of functions in $\mathcal{H}$ upper-bounded by 1 on $\mathcal{X}$. Let $m = |Range(f)|, \gamma > 0$, and $\alpha \leq \frac{\gamma^3}{16m}$. Then if $\mathcal{H}$ satisfies the $(\gamma, \frac{\gamma}{m})$-weak learning condition and $f$ is $\alpha$-approximately $\ell_2$ multicalibrated with respect to $\mathcal{H}_1$ and $P$, then $f$ has squared error*

$$\mathbb{E}_P[(f(x) - y)^2] \leq \mathbb{E}_P[(f^*(x) - y)^2] + 3\gamma,$$

*where $f^*(x) = \mathbb{E}_P[Y|x]$.*

**Definition F.3.** *For a probability measure $P(X, Y)$ and a predictor $f : \mathcal{X} \to [0, 1]$, let $\mathcal{H} \subset \mathbb{R}^{\mathcal{X} \times \mathcal{Y}}$ be a function class. We say that $f$ is $\alpha$-approximately $\ell_1$ multicalibrated w.r.t. $\mathcal{H}$ and $P$ if for all $h \in \mathcal{H}$:*

$$K_1(f, h, P) \tag{24}$$

$$= \int \left| \mathbb{E}_P \left[ h(X, Y)(Y - v) \big| f(X) = v \right] \right| dP_{f(X)}(v) \tag{25}$$

$$\leq \alpha. \tag{26}$$

**Lemma F.4** ( Roth [37]). *Suppose $P_S, P_T \in \mathcal{P}$ have the same conditional label distribution, and suppose $f$ is $\alpha$-approximately $\ell_1$ multicalibrated with respect to $P_S$ and $\mathcal{H}$. If $\mathcal{H}$ satisfies Equation 21, then $f$ is also $\alpha$-approximately $\ell_1$ multicalibrated with respect to $P_T$ and $\mathcal{H}$:*

**Lemma F.5.** *For a predictor $f : \mathcal{X} \to [0, 1]$ and a grouping function $h$ satisfying $\max_{x \in \mathcal{X}, y \in \mathcal{Y}} h(x, y)^2 \leq B$ where $B > 0$,*

$$\frac{1}{\sqrt{B}} K_2(f, h, P) \leq K_1(f, h, P) \leq \sqrt{K_2(f, h, P)}. \tag{27}$$

**Remark F.6.** *The lemma is extended from Roth [37]'s result for $B = 1$.*

*Proof.* First we prove $K_2(f, h, P) \leq \sqrt{B} K_1(f, h, P)$. For any $v \in [0, 1]$,

$$\left( \mathbb{E}_P \left[ h(X, Y)(Y - v) \big| f(X) = v \right] \right)^2 \tag{28}$$

$$= \left| \mathbb{E}_P \left[ h(X)(Y - v) \big| f(X) = v \right] \right| \cdot \left| \mathbb{E}_P \left[ h(X, Y)(Y - v) \big| f(X) = v \right] \right| \tag{29}$$

$$\leq \sqrt{\mathbb{E} \left[ h(X, Y)^2 \big| f(X) = v \right] \mathbb{E} \left[ (Y - v)^2 \big| f(X) = v \right]} \tag{30}$$

$$\cdot \left| \mathbb{E}_P \left[ h(X, Y)(Y - v) \big| f(X) = v \right] \right| \tag{31}$$

$$\leq \sqrt{B} \cdot \left| \mathbb{E}_P \left[ h(X, Y)(Y - v) \big| f(X) = v \right] \right|. \tag{32}$$

Equation 30 follows from the Cauchy-Schwarz inequality.

$$K_2(f, h, P) = \mathbb{E}_{v \sim P_{f(X)}} \left[ \left( \mathbb{E}_P \left[ h(X, Y)(Y - v) \big| f(X) = v \right] \right)^2 \right] \tag{33}$$

$$\leq \sqrt{B} \mathbb{E}_{v \sim P_{f(X)}} \left| \mathbb{E}_P \left[ h(X, Y)(Y - v) \big| f(X) = v \right] \right| \tag{34}$$

$$= \sqrt{B} K_1(f, h, P). \tag{35}$$

Then we prove $K_1(f, h, P) \leq \sqrt{BK_2(f, h, P)}$. Still from the Cauchy-Schwarz inequality:

$$K_1(f, h, P) = \mathbb{E}_{v \sim P_{f(X)}} \left| \mathbb{E}_P \left[ h(X, Y)(Y - v) \big| f(X) = v \right] \right| \tag{36}$$

$$\leq \sqrt{\mathbb{E}_{v \sim P_{f(X)}} [1^2] \mathbb{E}_{v \sim P_{f(X)}} \left[ \left( \mathbb{E}_P \left[ h(X, Y)(Y - v) \big| f(X) = v \right] \right)^2 \right]} \tag{37}$$

$$= \sqrt{K_2(f, h, P)}. \tag{38}$$

$$\square$$

**Theorem F.7** (Restatement of Theorem 2.4). *For a source measure $P_S(X, Y)$ and a set of probability measures $\mathcal{P}(X)$ containing $P_S(X)$, given a predictor $f : \mathcal{X} \to [0,1]$ with finite range $m := |Range(f)|$, consider a grouping function class $\mathcal{H}$ closed under affine transformation satisfying Assumption 2.3 with $\rho = \gamma/m$. If $f$ is $\frac{\gamma^6}{256m^2}$-approximately $\ell_2$ multicalibrated w.r.t $P_S$ and $\mathcal{H}$'s bounded subset $\mathcal{H}_1 := \{h \in \mathcal{H} : \max_{x \in \mathcal{X}} h(x)^2 \leq 1\}$, then for any target measure $P_T \in \mathcal{P}(X)P_S(Y|X)$,*

$$R_{P_T}(f) \leq R_{P_T}(f^*) + 3\gamma, \tag{39}$$

*where $f^*(x) = \mathbb{E}_{P_T}[Y|x]$ is the optimal regression function in each target distribution.*

*Proof.* For $|h(x)| \leq 1$ and $f \in [0,1]$, according to Lemma F.5,

$$K_2(f, h, P) \leq K_1(f, h, P) \leq \sqrt{K_2(f, h, P)}. \tag{40}$$

Since $K_2(f, h, P_S) \leq \frac{\gamma^6}{256m^2}$ for any $h \in H_1$,

$$K_1(f, h, P_S) \leq \frac{\gamma^3}{16m}. \tag{41}$$

With Lemma F.4, for any $h \in H_1$ and $P_T \in \mathcal{P}$, we have

$$K_1(f, h, P_T) \leq \frac{\gamma^3}{16m}. \tag{42}$$

Again, it follows from Lemma F.5 that:

$$K_2(f, h, P_T) \leq \frac{\gamma^3}{16m}. \tag{43}$$

With Lemma F.2, we have

$$\mathbb{E}_{P_T}[(f(x) - y)^2] \leq \mathbb{E}_{P_T}[(f^*(x) - y)^2] + 3\gamma, \tag{44}$$

which completes the proof. $\square$

## F.2 Multicalibration and Invariance under Concept Shift

**Theorem F.8** (Restatement of Theorem 3.1). *For a set of absolutely continuous probability measures $\mathcal{P}(X, Y)$ containing the source measure $P_S(X, Y)$, consider a predictor $f : \mathcal{X} \to [0,1]$. Assume the grouping function class $\mathcal{H}$ satisfies the following condition:*

$$\mathcal{H} \supset \left\{ h(x, y) = \frac{p(x, y)}{p_S(x, y)} \Big| P \in \mathcal{P}(X, Y) \right\}. \tag{45}$$

*If $f$ is $\alpha$-approximately $\ell_2$ multicalibrated w.r.t. $\mathcal{H}$ and $P_S$, then for any measure $P \in \mathcal{P}(X, Y)$,*

$$R_P(f) \leq \inf_{g:[0,1]\to[0,1]} R_P(g \circ f) + 2\sqrt{\alpha}. \tag{46}$$

*Proof.* For any $h(x, y) = p(x, y)/p_S(x, y)$ where $P \in \mathcal{P}$, since $f$ is $\alpha$-approximately $\ell_2$ multicalibrated, $K_2(f, h, P_S) \leq \alpha$. It follows from Lemma F.5 that $K_1(f, h, P_S) \leq \sqrt{\alpha}$.

$$K_1(f, h, P_S) = \int \left| \mathbb{E}_{P_S} \left[ h(X, Y)(Y - v) \big| f(X) = v \right] \right| dP_S(f^{-1}(v)) \tag{47}$$

$$= \int \left| \int \frac{p(x, y)}{p_S(x, y)} (y - v) p_S(x, y | f = v) d(x, y) \right| dP_S(f^{-1}(v)) \tag{48}$$

$$= \int \left| \int \frac{dP(f^{-1}(v))}{dP_S(f^{-1}(v))} (y - v) p(x, y | f = v) d(x, y) \right| dP_S(f^{-1}(v)) \tag{49}$$

$$= \int \frac{dP(f^{-1}(v))}{dP_S(f^{-1}(v))} \left| \int (y - v) p(x, y | f = v) d(x, y) \right| dP_S(f^{-1}(v)) \tag{50}$$

$$= \int \left| \mathbb{E}_P \left[ Y - v \big| f(X) = v \right] \right| dP(f^{-1}(v)) \tag{51}$$

$$= K_1(f, 1, P). \tag{52}$$

Thus, we have $K_1(f, 1, P) \leq \sqrt{\alpha}$ for any $P \in \mathcal{P}$. We will prove an equivalent form of $\ell_1$ calibration error:

$$K_1(f, 1, P) = \sup_{\eta:[0,1]\to[-1,1]} \int \eta(v)\mathbb{E}_P\left[Y - v\big|f(X) = v\right] dP_{f(X)}(v) \tag{53}$$

$$= \sup_{\eta:[0,1]\to[-1,1]} \mathbb{E}_P[\eta(f(X))(Y - f(X))]. \tag{54}$$

$K_1(f, 1, P) \leq \sup_{\eta:[0,1]\to[-1,1]} \mathbb{E}_P[\eta(f(X))(Y - f(X))]$ can be proved by taking $\eta(v) = 2\mathbb{I}[v > 0] - 1$. On the other hand,

$$\int \eta(v)\mathbb{E}_P\left[Y - v\big|f(X) = v\right] dP_{f(X)}(v) \leq \int \left|\eta(v)\right| \cdot \left|\mathbb{E}_P\left[Y - v\big|f(X) = v\right]\right| dP_{f(X)}(v) \tag{55}$$

$$\leq K_1(f, 1, P). \tag{56}$$

Actually the right hand side of Equation 53 resembles smooth calibration [3], which restricts $\eta$ to Lipschitz functions. Based on smooth calibration, Blasiok et al. [4] shows that approximately calibrated predictors cannot be improved much by post-processing. In the above we present a similar proof for $\ell_1$ calibration error.

For any $g : [0, 1] \to [0, 1]$, there exists $\eta : [0, 1] \to [-1, 1]$ such that $g(v) = v + \eta(v)$ for any $v \in [0, 1]$.

$$R_P(g \circ f) = \mathbb{E}_P\left[\left(Y - f(X) - \eta(f(X))\right)^2\right] \tag{57}$$

$$= \mathbb{E}_P\left[(Y - f(X))^2\right] - 2\mathbb{E}_P\left[(Y - f(X)\eta(f(X))\right] + \mathbb{E}_P\left[\eta(f(X))^2\right] \tag{58}$$

$$\geq R_P(f) - \sup_{\eta':[0,1]\to[-1,1]} 2\mathbb{E}[\eta'(f(X))(Y - f(X))] \tag{59}$$

$$= R_P(f) - 2K_1(f, 1, P) \tag{60}$$

$$\geq R_P(f) - 2\sqrt{\alpha}. \tag{61}$$

$\square$

**Theorem F.9** (Restatement of Theorem 3.4). *Assume samples are drawn from an environment $e \in \mathcal{E}$ with a prior $P_S(e)$ such that $\sum_{e \in \mathcal{E}} P_S(e) = 1$ and $P_S(e) > 0$. The overall population satisfies $P_S(X, Y) = \sum_{e \in \mathcal{E}} P_e(X, Y)P_S(e)$ where $P_e(X, Y)$ is the environment-specific absolutely continuous measure. For a representation $\Phi \in \sigma(X)$ over features, define a function class $\mathcal{H}$ as:*

$$\mathcal{H} := \left\{h(x, y) = \frac{p_e(x, y)}{p_S(x, y)}\Big|e \in \mathcal{E}\right\}. \tag{62}$$

*1. If there exists a bijection $g^* : supp(\Phi) \to [0, 1]$ such that $g^* \circ \Phi$ is $\alpha$-approximately $\ell_2$ multicalibrated w.r.t. $\mathcal{H}$ and $P_S$, then for any $e \in \mathcal{E}$,*

$$R_{P_e}(g^* \circ \Phi) \leq \inf_{g:supp(\Phi)\to[0,1]} R_{P_e}(g \circ \Phi) + 2\sqrt{\alpha}. \tag{63}$$

*2. For $\Phi \in \sigma(X)$, if there exists $g^* : supp(\Phi) \to [0, 1]$ such that Equation 63 is satisfied for any $e \in \mathcal{E}$, then $g^* \circ \Phi$ is $\sqrt{2/D}\alpha^{1/4}$-approximately $\ell_2$ multicalibrated w.r.t. $\mathcal{H}$ and $P_S$, where $D = \min_{e \in \mathcal{E}} P_S(e)$.*

*Proof.* We first prove statement 1.

For any $g : supp(\Phi) \to [0, 1]$, since $g^*$ is a bijection, $g \circ \Phi = (g \circ g^{*-1})(g^* \circ \Phi)$ where $g \circ g^{*-1} \in [0, 1]^{[0,1]}$. Since $g^* \circ \Phi$ is $\alpha$-approximately $\ell_2$ multicalibrated, it follows from Theorem F.8 that for any $e \in \mathcal{E}$,

$$R_{P_e}(g^* \circ \Phi) \leq R_{P_e}\left((g \circ g^{*-1})(g^* \circ \Phi)\right) + 2\sqrt{\alpha} \tag{64}$$

$$= R_{P_e}(g \circ \Phi) + 2\sqrt{\alpha}. \tag{65}$$

Then we give a proof of statement 2, which is inspired by Blasiok et al. [4].

For simplicity let $f^* := g^* \circ \Phi$. For any $e \in \mathscr{E}$ and any $\eta : [0,1] \to [-1,1]$, define $\beta := E_{P_e}[(Y - f^*(X))\eta(f^*(X))] \in [-1,1]$. Construct $\kappa(v) := \text{proj}_{[0,1]}(v + \beta\eta(v))$, where $\text{proj}_{[0,1]}(\cdot) = \max\{0, \min\{1, \cdot\}\}$.

$$R_{P_e}(\kappa(f^*)) = \mathbb{E}_{P_e}\left[\left(Y - \kappa(f^*(X))\right)^2\right] \tag{66}$$

$$\leq \mathbb{E}_{P_e}\left[\left(Y - f^*(X) - \beta\eta(f^*(X))\right)^2\right] \tag{67}$$

$$= \mathbb{E}_{P_e}\left[(Y - f^*(X))^2\right] - 2\beta^2 + \beta^2\mathbb{E}_{P_e}[\eta(f^*(x))^2] \tag{68}$$

$$\leq R_{P_e}(f^*) - 2\beta^2 + \beta^2 \tag{69}$$

$$= R_{P_e}(f^*) - \beta^2. \tag{70}$$

Rearranging the inequality above gives:

$$\left(E_{P_e}[(Y - f^*(X))\eta(f^*(X))]\right)^2 = \beta^2 \leq R_{P_e}(f^*) - R_{P_e}(\kappa(f^*)). \tag{71}$$

Since $\kappa \circ g^* \in \text{supp}(\Phi) \to [0,1]$, it follows from Equation 63 that:

$$R_{P_e}(f^*) - R_{P_e}(\kappa(f^*)) = R_{P_e}(g^* \circ \Phi) - R_{P_e}((\kappa \circ g^*) \circ \Phi) \leq 2\sqrt{\alpha}. \tag{72}$$

Combining Equation 71 and Equation 72 gives $E_{P_e}[(Y - f^*(X))\eta(f^*(X))] \leq \sqrt{2}\alpha^{1/4}$ for any $\eta : [0,1] \to [-1,1]$. From Equation 53, it follows that:

$$K_1(f^*, 1, P_e) = \sup_{\eta:[0,1]\to[-1,1]} \mathbb{E}_{P_e}[\eta(f^*(X))(Y - f^*(X))] \leq \sqrt{2}\alpha^{1/4}. \tag{73}$$

From Equation 52, $K_1(f^*, h, P_S) \leq \sqrt{2}\alpha^{1/4}$ for any $h \in \mathcal{H}$. Further, for any $h \in \mathcal{H}$,

$$h(x, y) = \frac{p_e(x, y)}{p_S(x, y)} \tag{74}$$

$$= \frac{p_e(x, y)}{\sum_{e' \in \mathscr{E}} p_{e'}(x, y)P_S(e')} \tag{75}$$

$$\leq \frac{p_e(x, y)}{p_e(x, y)P_S(e)} \tag{76}$$

$$\leq \frac{1}{D}. \tag{77}$$

By Lemma F.5, it follows that $K_2(f^*, h, P_S) \leq \sqrt{1/D} \cdot K_1(f^*, h, P_S) \leq \sqrt{2/D}\alpha^{1/4}$. $\quad\square$

### F.3 Structure of Grouping Function Classes

In this subsection, we focus on Euclidean space where $\mathcal{X} \subset \mathbb{R}^d$ is compact and measurable for some $d \in Z^+$ and $\mathcal{Y} = [0,1]$. Grouping functions are assumed to be continuous, i.e., $h \in C(\mathcal{X} \times \mathcal{Y})$. We consider absolutely continuous probability measures with continuous density functions.

**Proposition F.10** (Restatement of Proposition 4.1). *Consider an absolutely continuous probability measure $P_S(X, Y)$ and a predictor $f : \mathcal{X} \to [0,1]$, define the maximal grouping function class that $f$ is multicalibrated with respect to:*

$$\mathcal{H}_f := \{h \in C(\mathcal{X} \times \mathcal{Y}) : K_2(f, h, P_S) = 0\}. \tag{78}$$

*Then $\mathcal{H}_f$ is a linear space.*

*Particularly for $f_\Phi(x) = \mathbb{E}[Y|\Phi(x)]$ where $\Phi : \mathbb{R}^d \to \mathbb{R}^{d_\Phi}, d_\Phi \in Z^+$ is a measurable function, we abbreviate $\mathcal{H}_{f_\Phi}$ with $\mathcal{H}_\Phi$. Then any finite subset $\mathcal{G} \subset \mathcal{H}_\Phi$ implies $\text{span}\{1, \mathcal{G}\} \subset \mathcal{H}_\Phi$, where $1$ denotes a constant function.*

*Proof.* For any $\gamma_1, \gamma_2 \in \mathbb{R}$ and any $h_1, h_2 \in \mathcal{H}_f$, $\gamma_1 h_1 + \gamma_2 h_2 \in C(\mathcal{X} \times \mathcal{Y})$.

$$K_1(f, \gamma_1 h_1 + \gamma_2 h_2, P_S) = \int \left| \mathbb{E}\left[ (\gamma_1 h_1(X,Y) + \gamma_2 h_2(X,Y))(Y-v) \big| f(X) = v \right] \right| dP_S(f^{-1}(v)) \tag{79}$$

$$\leq \int \left| \mathbb{E}\left[ \gamma_1 h_1(X,Y)(Y-v) \big| f(X) = v \right] \right| dP_S(f^{-1}(v)) \tag{80}$$

$$+ \int \left| \mathbb{E}\left[ \gamma_2 h_2(X,Y)(Y-v) \big| f(X) = v \right] \right| dP_S(f^{-1}(v)) \tag{81}$$

$$= \gamma_1 K_1(f, h_1, P_S) + \gamma_2 K_1(f, h_2, P_S) \tag{82}$$

$$= 0. \tag{83}$$

According to Lemma F.5, $K_1(f, h, P_S) = 0$ is equivalent as $K_2(f, h, P_S) = 0$ for bounded $h$. Thus, $K_2(f, \gamma_1 h_1 + \gamma_2 h_2, P_S) = 0$ which implies $\gamma_1 h_1 + \gamma_2 h_2 \in \mathcal{H}_f$. Now we finishes the proof that $\mathcal{H}_f$ is a linear space.

For $f_\Phi(x) = \mathbb{E}[Y|\Phi(x)]$,

$$\mathbb{E}[Y|f_\Phi(X)] = \mathbb{E}\left[ \mathbb{E}[Y|f_\Phi(X), \Phi(X)] \big| f_\Phi(X) \right] \tag{84}$$

$$= \mathbb{E}\left[ \mathbb{E}[Y|\Phi(X)] \big| f_\Phi(X) \right] \tag{85}$$

$$= \mathbb{E}[f_\Phi(X)|f_\Phi(X)] \tag{86}$$

$$= f_\Phi(X). \tag{87}$$

Thus, $K_2(f_\Phi, 1, P_S) = 0$ which implies $1 \in \mathcal{H}_\Phi$. Since $\mathcal{H}_\Phi$ is a linear space, $\mathcal{G} \subset \mathcal{H}_\Phi$ implies $\text{span}\{1, \mathcal{G}\} \subset \mathcal{H}_\Phi$. $\qquad\square$

**Lemma F.11.** *For any absolutely continuous probability measure $P(X, Y)$ with a continuous density function $p$, and any grouping function $h \in C(\mathcal{X} \times \mathcal{Y})$, there exists $\gamma \neq 0$ and $\rho \in \mathbb{R}$ such that $\gamma h + \rho \in C(\mathcal{X} \times \mathcal{Y})$, and it is density ratio between some absolutely continuous probability measure $P'$ and $P$, i.e., $dP'(x,y) = (\gamma h(x,y) + \rho)dP(x,y)$, where $P'$ also has a continuous density function.*

*Proof.* Since $h$ is bounded, there exists $\rho' \in \mathbb{R}$ such that $h(x,y) + \rho' > 0$ for any $x \in \mathcal{X}, y \in \mathcal{Y}$. Define:

$$\gamma = \frac{1}{\int (h(x,y) + \rho')dP(x,y)}. \tag{88}$$

$$\rho = \frac{\rho'}{\int (h(x,y) + \rho')dP(x,y)}. \tag{89}$$

$\gamma h + \rho$ is still continuous.

We have $\gamma h(x,y) + \rho = \gamma(h(x,y) + \rho') > 0$ for any $x \in \mathcal{X}, y \in \mathcal{Y}$.

$$\int dP'(x,y) = \int (\gamma h(x,y) + \rho)dP(x,y) \tag{90}$$

$$= \gamma \int (h(x,y) + \rho')dP(x,y) \tag{91}$$

$$= 1. \tag{92}$$

Thus, $P'(X, Y)$ is an absolutely continuous probability measure.

Its density function $p' = (\gamma h + \rho)p$ is continuous. $\qquad\square$

**Theorem F.12.** *Consider an absolutely continuous probability measure $P_S(X, Y)$ and a predictor $f_\Phi(x) = \mathbb{E}_{P_S}[Y|\Phi(x)]$ where $\Phi : \mathbb{R}^d \to \mathbb{R}^{d_\Phi}, d_\Phi \in Z^+$ is a measurable function. We abbreviate $\mathcal{H}_{f_\Phi}$ with $\mathcal{H}_\Phi$.*

$$\mathcal{H}_\Phi = \{h \in C(\mathcal{X} \times \mathcal{Y}) : Cov_{P_S}\left[h(X,Y), Y|f_\Phi = v\right] = 0 \text{ for almost every } v \in [0,1]\} \tag{93}$$

$$= span\left\{ \frac{p(x,y)}{p_S(x,y)} : \mathbb{E}_P[Y|f_\Phi] = \mathbb{E}_{P_S}[Y|f_\Phi] \text{ almost surely, } p \text{ is continuous} \right\}. \tag{94}$$

*Proof.* First we prove:

$$\mathcal{H}_\Phi = \{h \in C(\mathcal{X} \times \mathcal{Y}) : \mathrm{Cov}_{P_S}[h(X,Y)|f_\Phi = v] = 0 \text{ for almost every } v \in [0,1]\}. \tag{95}$$

For each $v \in [0,1]$ and any $h \in C(\mathcal{X} \times \mathcal{Y})$,

$$\mathbb{E}_{P_S}\left[h(X,Y)(Y-v)\big|f_\Phi = v\right] = \mathbb{E}_{P_S}\left[h(X,Y)Y\big|f_\Phi = v\right] - v\mathbb{E}_{P_S}\left[h(X,Y)\big|f_\Phi = v\right] \tag{96}$$

$$= \mathbb{E}_{P_S}\left[h(X,Y)Y\big|f_\Phi = v\right] \tag{97}$$

$$- \mathbb{E}_{P_S}\left[h(X,Y)\big|f_\Phi = v\right]\mathbb{E}[Y|f_\Phi = v] \tag{98}$$

$$= \mathrm{Cov}_{P_S}[h(X,Y),Y|f_\Phi = v]. \tag{99}$$

Equation 98 follows from Equation 87.

For any $h \in C(\mathcal{X} \times \mathcal{Y})$,

$$h \in \mathcal{H}_\Phi \Leftrightarrow K_2(f_\Phi, h, P_S) = 0 \tag{100}$$

$$\Leftrightarrow \int \left(\mathbb{E}_{P_S}\left[h(X,Y)(Y-v)\big|f_\Phi = v\right]\right)^2 dP_S(f_\Phi^{-1}(v)) \tag{101}$$

$$\Leftrightarrow \mathbb{E}_{P_S}\left[h(X,Y)(Y-v)\big|f_\Phi = v\right] = 0 \text{ for almost every } v \in [0,1] \tag{102}$$

$$\Leftrightarrow \mathrm{Cov}_{P_S}[h(X,Y),Y|f_\Phi = v] = 0 \text{ for almost every } v \in [0,1]. \tag{103}$$

Next, we prove

$$\mathcal{H}_\Phi \supset \mathrm{span}\left\{\frac{p(x,y)}{p_S(x,y)} : \mathbb{E}_P[Y|f_\Phi] = \mathbb{E}_{P_S}[Y|f_\Phi] \text{ almost surely, } p \text{ is continuous}\right\}. \tag{104}$$

This is equivalent to saying that for any absolutely continuous probability measure $P$ satisfying $\mathbb{E}_P[Y|f_\Phi] = \mathbb{E}_{P_S}[Y|f_\Phi]$ almost surely, $\mathrm{Cov}_{P_S}[p(X,Y)/p_S(X,Y), Y|f_\Phi = v] = 0$ for almost every $v \in [0,1]$.

$$\mathrm{Cov}_{P_S}\left[\frac{p(X,Y)}{p_S(X,Y)}, Y\big|f_\Phi = v\right] \tag{105}$$

$$= \mathbb{E}_{P_S}\left[\frac{p(X,Y)}{p_S(X,Y)}Y\big|f_\Phi = v\right] - \mathbb{E}_{P_S}\left[\frac{p(X,Y)}{p_S(X,Y)}\big|f_\Phi = v\right]\mathbb{E}_{P_S}\left[Y\big|f_\Phi = v\right] \tag{106}$$

$$= \int \frac{dP(f_\Phi^{-1}(v))}{dP_S(f_\Phi^{-1}(v))}\frac{p(x,y|f_\Phi = v)}{p_S(x,y|f_\Phi = v)}y \cdot dP_S(x,y|f_\Phi = v) \tag{107}$$

$$- \mathbb{E}_{P_S}\left[Y\big|f_\Phi = v\right] \cdot \int \frac{dP(f_\Phi^{-1}(v))}{dP_S(f_\Phi^{-1}(v))}\frac{p(x,y|f_\Phi = v)}{p_S(x,y|f_\Phi = v)} \cdot dP_S(x,y|f_\Phi = v) \tag{108}$$

$$= \mathbb{E}_P\left[Y\big|f_\Phi = v\right]\frac{dP(f_\Phi^{-1}(v))}{dP_S(f_\Phi^{-1}(v))} - \mathbb{E}_{P_S}\left[Y\big|f_\Phi = v\right]\frac{dP(f_\Phi^{-1}(v))}{dP_S(f_\Phi^{-1}(v))} \tag{109}$$

$$= \left[\mathbb{E}_P\left[Y\big|f_\Phi = v\right] - \mathbb{E}_{P_S}\left[Y\big|f_\Phi = v\right]\right]\frac{dP(f_\Phi^{-1}(v))}{dP_S(f_\Phi^{-1}(v))} \tag{110}$$

$$= 0 \quad \text{almost surely.} \tag{111}$$

Next, we prove

$$\mathcal{H}_\Phi \subset \mathrm{span}\left\{\frac{p(x,y)}{p_S(x,y)} : \mathbb{E}_P[Y|f_\Phi] = \mathbb{E}_{P_S}[Y|f_\Phi] \text{ almost surely, } p \text{ is continuous}\right\}. \tag{112}$$

By Lemma F.11, any grouping function $h \in \mathcal{H}_\Phi$ could be rewritten as $h(x,y) = \gamma p(x,y)/p_S(x,y) + \rho$ for some continuous density functions $p$ and $\gamma \neq 0$. Thus, we just need to prove the statement that

$\text{Cov}_{P_S}[\gamma p(X,Y)/p_S(X,Y) + \rho, Y | f_\Phi = v] = 0$ implies $\mathbb{E}_P[Y|f_\Phi = v] = \mathbb{E}_{P_S}[Y|f_\Phi = v]$.

$$\text{Cov}_{P_S}\left[\gamma \frac{p(X,Y)}{p_S(X,Y)} + \rho, Y \Big| f_\Phi = v\right] \tag{113}$$

$$= \gamma \mathbb{E}_{P_S}\left[\frac{p(X,Y)}{p_S(X,Y)} Y \Big| f_\Phi = v\right] + \rho E_{P_S}\left[Y \Big| f_\Phi = v\right] \tag{114}$$

$$- \gamma E_{P_S}\left[\frac{p(X,Y)}{p_S(X,Y)} \Big| f_\Phi = v\right] E_{P_S}\left[Y \Big| f_\Phi = v\right] - \rho E_{P_S}\left[Y \Big| f_\Phi = v\right] \tag{115}$$

$$= \gamma \mathbb{E}_{P_S}\left[\frac{p(X,Y)}{p_S(X,Y)} Y \Big| f_\Phi = v\right] - \gamma E_{P_S}\left[\frac{p(X,Y)}{p_S(X,Y)} \Big| f_\Phi = v\right] E_{P_S}\left[Y \Big| f_\Phi = v\right] \tag{116}$$

$$= \gamma \text{Cov}_{P_S}\left[\frac{p(X,Y)}{p_S(X,Y)}, Y \Big| f_\Phi = v\right] \tag{117}$$

$$= \gamma \left[\mathbb{E}_P\left[Y \Big| f_\Phi = v\right] - \mathbb{E}_{P_S}\left[Y \Big| f_\Phi = v\right]\right] \frac{dP(f_\Phi^{-1}(v))}{dP_S(f_\Phi^{-1}(v))}. \tag{118}$$

Equation 118 follows from Equation 110. So we have $\mathbb{E}_P[Y|f_\Phi = v] = \mathbb{E}_{P_S}[Y|f_\Phi = v]$ if $\text{Cov}_{P_S}[\gamma p(X,Y)/p_S(X,Y) + \rho, Y|f_\Phi = v] = 0$. $\qquad\square$

**Corollary F.13** (Restatement of Theorem 4.2). *Consider an absolutely continuous probability measure $P_S(X,Y)$ and a calibrated predictor $f$.*

$$\mathcal{H}_f = \{h \in C(\mathcal{X} \times \mathcal{Y}) : Cov_{P_S}[h(X,Y), Y|f = v] = 0 \text{ for almost every } v \in [0,1]\} \tag{119}$$

$$= span\left\{\frac{p(x,y)}{p_S(x,y)} : \mathbb{E}_P[Y|f] = \mathbb{E}_{P_S}[Y|f] \text{ almost surely, } p \text{ is continuous}\right\}. \tag{120}$$

**Remark F.14.** $\mathbb{E}_P[Y|f] = \mathbb{E}_{P_S}[Y|f]$ *almost surely implies* $\mathbb{E}_P[Y|f] = f$*, which is equivalent as* $R_P(f) = \inf_{g:[0,1]\to[0,1]} R_P(g \circ f)$*, since we adopt square error.*

*Proof.* Since $f$ is calibrated, we have $\mathbb{E}_{P_S}[Y|f] = f$. Take $\Phi(x) = f(x)$.

$$f_\Phi(x) = \mathbb{E}_{P_S}[Y|\Phi(x)] = \mathbb{E}_{P_S}[Y|f(x)] = f(x). \tag{121}$$

Apply Theorem F.12 and the proof is complete. $\qquad\square$

**Theorem F.15** (Restatement of Theorem 4.3 (first part)). *Consider an absolutely continuous probability measure $P_S(X,Y)$ and a predictor $f_\Phi(x) = \mathbb{E}_{P_S}[Y|\Phi(x)]$ where $\Phi : \mathbb{R}^d \to \mathbb{R}^{d_\Phi}, d_\Phi \in Z^+$ is a measurable function. $\mathcal{H}_\Phi$ can be decomposed as $\mathcal{H}_\Phi = \mathcal{H}_{1,\Phi} + \mathcal{H}_{2,\Phi}$.*

$$\mathcal{H}_{1,\Phi} := \{h \in C(\mathcal{X} \times \mathcal{Y}) : Cov_{P_S}[h(\Phi,Y), Y|f_\Phi = v] = 0 \text{ for almost every } v \in [0,1]\} \tag{122}$$

$$= span\left\{\frac{p(\Phi,y)}{p_S(\Phi,y)} : \mathbb{E}_P[Y|f_\Phi] = \mathbb{E}_{P_S}[Y|f_\Phi] \text{ almost surely, } p \text{ is continuous}\right\}. \tag{123}$$

$$\mathcal{H}_{2,\Phi} := \{h \in C(\mathcal{X} \times \mathcal{Y}) : \mathbb{E}[h(X,Y)|\Phi,Y] \equiv C_h \;\; \forall \Phi, Y\} \tag{124}$$

$$= span\left\{\frac{p(x|\Phi,y)}{p_S(x|\Phi,y)} : \; p \text{ is continuous}\right\}. \tag{125}$$

**Remark F.16.** $\mathcal{H}_{1,\Phi}$ *contains functions defined on $supp(\Phi) \times \mathcal{Y}$ which can be rewritten as functions on $\mathcal{X} \times \mathcal{Y}$ by variable substitution. Thus, $\mathcal{H}_{1,\Phi}$ is still a set of grouping functions. For $\mathcal{H}_{2,\Phi}$, $C_h$ is a constant depending on $h$.*

*Proof.* First we prove $\mathcal{H}_\Phi \subset \mathcal{H}_{1,\Phi} + \mathcal{H}_{2,\Phi}$.

For any $h_1 \in \mathcal{H}_{1,\Phi}$ and $h_2 \in \mathcal{H}_{2,\Phi}$,

$$\text{Cov}_{P_S}\left[h_1(\Phi, Y) + h_2(X, Y), Y | f_\Phi = v\right] \tag{126}$$

$$= \text{Cov}_{P_S}\left[h_1(\Phi, Y), Y | f_\Phi = v\right] + \text{Cov}_{P_S}\left[h_2(X, Y), Y | f_\Phi = v\right] \tag{127}$$

$$= \text{Cov}_{P_S}\left[h_2(X, Y), Y | f_\Phi = v\right] \tag{128}$$

$$= \mathbb{E}_{P_S}\left[h_2(X, Y)Y | f_\Phi = v\right] - \mathbb{E}_{P_S}\left[h_2(X, Y) | f_\Phi = v\right] \mathbb{E}_{P_S}\left[Y | f_\Phi = v\right] \tag{129}$$

$$= \mathbb{E}_{P_S}\left[\mathbb{E}_{P_S}\left[h_2(X, Y)Y | \Phi, Y\right] \big| f_\Phi = v\right] - \mathbb{E}_{P_S}\left[\mathbb{E}_{P_S}\left[h_2(X, Y) | \Phi, Y\right] \big| f_\Phi = v\right] \mathbb{E}_{P_S}\left[Y | f_\Phi = v\right] \tag{130}$$

$$= \mathbb{E}_{P_S}\left[\mathbb{E}_{P_S}\left[h_2(X, Y) | \Phi, Y\right] Y \big| f_\Phi = v\right] - \mathbb{E}_{P_S}\left[\mathbb{E}_{P_S}\left[h_2(X, Y) | \Phi, Y\right] \big| f_\Phi = v\right] \mathbb{E}_{P_S}\left[Y | f_\Phi = v\right] \tag{131}$$

$$= C_{h_2} E_{P_S}\left[Y | f_\Phi = v\right] - C_{h_2} E_{P_S}\left[Y | f_\Phi = v\right] \tag{132}$$

$$= 0. \tag{133}$$

Next we prove $\mathcal{H}_\Phi \supset \mathcal{H}_{1,\Phi} + \mathcal{H}_{2,\Phi}$.

For any $h \in \mathcal{H}_\Phi$, let

$$h_1(\Phi(x), y) = \mathbb{E}_{P_S}[h(X, Y) | \Phi = \Phi(x), Y = y]. \tag{134}$$
$$h_2(x, y) = h(x, y) - \mathbb{E}_{P_S}[h(X, Y) | \Phi = \Phi(x), Y = y]. \tag{135}$$

Then we have

$$\text{Cov}_{P_S}\left[h_1(\Phi, Y), Y | f_\Phi = v\right] \tag{136}$$

$$= \mathbb{E}_{P_S}\left[h_1(\Phi, Y)Y | f_\Phi = v\right] - \mathbb{E}_{P_S}\left[h_1(\Phi, Y) | f_\Phi = v\right] \mathbb{E}_{P_S}\left[Y | f_\Phi = v\right] \tag{137}$$

$$= \mathbb{E}_{P_S}\left[\mathbb{E}_{P_S}[h(X, Y) | \Phi, Y] Y \big| f_\Phi = v\right] - \mathbb{E}_{P_S}\left[\mathbb{E}_{P_S}[h(X, Y) | \Phi, Y] \big| f_\Phi = v\right] \mathbb{E}_{P_S}\left[Y | f_\Phi = v\right] \tag{138}$$

$$= \mathbb{E}_{P_S}\left[h(X, Y)Y | f_\Phi = v\right] - \mathbb{E}_{P_S}\left[h(X, Y) | f_\Phi = v\right] \mathbb{E}_{P_S}\left[Y | f_\Phi = v\right] \tag{139}$$

$$= \text{Cov}_{P_S}\left[h(X, Y), Y | f_\Phi = v\right] \tag{140}$$

$$= 0 \quad \text{for almost every } v \in [0, 1]. \tag{141}$$

Thus, $h_1(\Phi(x), y) \in \mathcal{H}_{1,\Phi}$.

$$\mathbb{E}_{P_S}[h_2(X, Y) | \Phi, Y] = \mathbb{E}_{P_S}\left[h(X, Y) - \mathbb{E}_{P_S}[h(X, Y) | \Phi, Y] \big| \Phi, Y\right] \tag{142}$$

$$= \mathbb{E}_{P_S}[h(X, Y) | \Phi, Y] - \mathbb{E}_{P_S}[h(X, Y) | \Phi, Y] \tag{143}$$

$$= 0. \tag{144}$$

Thus, $h_2(x, y) \in \mathcal{H}_{2,\Phi}$. Following a similar proof of Theorem F.12, we have

$$\mathcal{H}_{1,\Phi} = \text{span}\left\{\frac{p(\Phi, y)}{p_S(\Phi, y)} : \mathbb{E}_P[Y | f_\Phi] = \mathbb{E}_{P_S}[Y | f_\Phi] \text{ almost surely, } p \text{ is continuous}\right\}. \tag{145}$$

Next, we prove $\mathcal{H}_{2,\Phi} \supset \text{span}\left\{p(x | \Phi, y) / p_S(x | \Phi, y) : p \text{ is continuous}\right\}$.

This is equivalent to saying that $p(x | \Phi, y) / p_S(x | \Phi, y) \in \mathcal{H}_{2,\Phi}$ for any continuous density function $p$.

$$\mathbb{E}_{P_S}\left[\frac{p(X | \Phi, Y)}{p_S(X | \Phi, Y)} \bigg| \Phi, Y\right] = \int \frac{p(x | \Phi, Y)}{p_S(x | \Phi, Y)} dP_S(x | \Phi, Y) \tag{146}$$

$$= \int dP(x | \Phi, Y) \tag{147}$$

$$= 1. \tag{148}$$

Thus, we have $p(x | \Phi, y) / p_S(x | \Phi, y) \in \mathcal{H}_{2,\Phi}$.

Next, we prove $\mathcal{H}_{2,\Phi} \subset \text{span}\left\{p(x | \Phi, y) / p_S(x | \Phi, y) : p \text{ is continuous}\right\}$.

By Lemma F.11, any grouping function $h \in \mathcal{H}_{2,\Phi}$ could be rewritten as $h_2(x, y) = \gamma p(x, y) / p_S(x, y) + \rho$ for some continuous density function $p$ and $\gamma \neq 0$. Thus, we just need to prove the statement that $\mathbb{E}_{P_S}[\gamma p(X, Y) / p_S(X, Y) + \rho | \Phi, Y] \equiv C_{h_2}$ implies $p(X, Y) / p_S(X, Y) \equiv$

$p(X|\Phi,Y)/p_S(X|\Phi,Y)$.

$$\mathbb{E}_{P_S}\left[\gamma\frac{p(X,Y)}{p_S(X,Y)}+\rho\Big|\Phi,Y\right]=\mathbb{E}_{P_S}\left[\gamma\frac{p(X|\Phi,Y)}{p_S(X|\Phi,Y)}\frac{p(\Phi,Y)}{p_S(\Phi,Y)}+\rho\Big|\Phi,Y\right] \tag{149}$$

$$=\gamma\frac{p(\Phi,Y)}{p_S(\Phi,Y)}\mathbb{E}_{P_S}\left[\frac{p(X|\Phi,Y)}{p_S(X|\Phi,Y)}\Big|\Phi,Y\right]+\rho \tag{150}$$

$$=\gamma\frac{p(\Phi,Y)}{p_S(\Phi,Y)}\mathbb{E}_{P}\left[1\Big|\Phi,Y\right]+\rho \tag{151}$$

$$=\gamma\frac{p(\Phi,Y)}{p_S(\Phi,Y)}+\rho. \tag{152}$$

Thus, we have $p(\Phi,Y)/p_S(\Phi,Y) \equiv (C_{h_2}-\rho)/\gamma$ which is a constant. Since $\mathbb{E}_{P_S}[p(\Phi,Y)/p_S(\Phi,Y)]=1$, we have $p(\Phi,Y)/p_S(\Phi,Y)\equiv 1$.

$$\frac{p(X,Y)}{p_S(X,Y)}=\frac{p(X|\Phi,Y)}{p_S(X|\Phi,Y)}\frac{p(\Phi,Y)}{p_S(\Phi,Y)} \tag{153}$$

$$=\frac{p(X|\Phi,Y)}{p_S(X|\Phi,Y)}. \tag{154}$$

$\square$

**Lemma F.17** (Theorem 3.2 from Globus-Harris et al. [13]). *If $f$ is calibrated and there exists an $h(x)$ such that*

$$\mathbb{E}[(f(X)-Y)^2-(h(X)-Y)^2|f(X)=v]\geq\alpha, \tag{155}$$

*then:*

$$\mathbb{E}[h(X)(Y-v)|f(X)=v]\geq\frac{\alpha}{2}. \tag{156}$$

**Proposition F.18** (Restatement of Theorem 4.3 (second part)). *If a predictor $f$ is multicalibrated with $\mathcal{H}_{1,\Phi}$, then $R_{P_S}(f)\leq R_{P_S}(f_\Phi)$.*

*Proof.* We prove by contradiction. If $R_{P_S}(f)>R_{P_S}(f_\Phi)$, then

$$\int\mathbb{E}\left[(f(X)-Y)^2-(f_\Phi(X)-Y)^2|f(X)=v\right]dP_S(f^{-1}(v))>0. \tag{157}$$

Let $\alpha_v=\mathbb{E}\left[(f(X)-Y)^2-(h(X)-Y)^2|f(X)=v\right]$.

Since $f$ is multicalibrated with $\mathcal{H}_{1,\Phi}$, $f$ is calibrated. It follows from Lemma F.17:

$$\mathbb{E}[f_\Phi(X)(Y-v)|f(X)=v]\geq\frac{\alpha_v}{2}. \tag{158}$$

Then,

$$K_1(f,f_\Phi,P_S)=\int\left|E[f_\Phi(X)(Y-v)|f(X)=v]\right|dP_S(f^{-1}(v)) \tag{159}$$

$$\geq\int\alpha_v dP_S(f^{-1}(v)) \tag{160}$$

$$>0. \tag{161}$$

From Lemma F.5, we have $K_2(f,f_\Phi,P_S)\geq K_1(f,f_\Phi,P_S)^2>0$. Since $f_\Phi\in\mathcal{H}_{1,\Phi}$, it contradicts with the fact that $f$ is multicalibrated with $\mathcal{H}_{1,\Phi}$. $\square$

**Proposition F.19** (Restatement of Theorem 4.3 (third part)). *$f_\Phi$ is an invariant predictor elicited by $\Phi$ across a set of environments $\mathcal{E}$ where $P_e(\Phi,Y)=P_S(\Phi,Y)$ for any $e\in\mathcal{E}$. If a predictor $f$ is multicalibrated with $\mathcal{H}_{2,\Phi}$, then $f$ is also an invariant predictor across $\mathcal{E}$ elicited by some representation.*

*Proof.* Since $P_e(\Phi, Y) = P_S(\Phi, Y)$, we have $\mathbb{E}_{P_e}[Y|\Phi] = \mathbb{E}_{P_S}[Y|\Phi] = f_\Phi$ for every $e \in \mathscr{E}$. Thus,

$$R_{P_e}(f_\Phi) = \inf_{g \in \mathcal{G}} R_{P_e}(g \circ \Phi). \tag{162}$$

This implies $f_\Phi$ is an invariant predictor elicited by $\Phi$ across $\mathscr{E}$.

For any $e \in \mathscr{E}$, we have:

$$\frac{p_e(x,y)}{p_S(x,y)} = \frac{p_e(\Phi,y)}{p_S(\Phi,y)} \frac{p_e(x|Phi,y)}{p_S(x|\Phi,y)} = \frac{p_e(x|\Phi,y)}{p_S(x|\Phi,y)} \in \mathcal{H}_{2,\Phi}. \tag{163}$$

Since $f$ is multicalibrated with $\mathcal{H}_{2,\Phi}$, it follows from Theorem F.8:

$$R_{P_e}(f) = \inf_{g:[0,1]\to[0,1]} R_{P_e}(g \circ f). \tag{164}$$

This implies that $f$ is an invariant predictor across $\mathscr{E}$ elicited by $f$. $\square$

**Proposition F.20** (Restatement of Proposition 4.5). *Consider $X \in \mathbb{R}^d$ which could be sliced as $X = (\Phi, \Psi)^T$ and $\Phi = (\Lambda, \Omega)^T$. Define $\mathcal{H}'_{1,\Phi} := \{h(\Phi(x)) \in C(\mathcal{X} \times \mathcal{Y})\}$, with $\mathcal{H}'_{1,\Phi} \subset \mathcal{H}_{1,\Phi}$. $\mathcal{H}'_{1,X}$ and $\mathcal{H}'_{1,\Lambda}$ are similarly defined. We have:*

*1. $\mathcal{H}'_{1,X} \supset \mathcal{H}'_{1,\Phi} \supset \mathcal{H}'_{1,\Lambda} \supset \mathcal{H}'_{1,\emptyset} = \{C\}$.*

*2. $\{C\} = \mathcal{H}_{2,X} \subset H_{2,\Phi} \subset \mathcal{H}_{2,\Lambda} \subset \mathcal{H}_{2,\emptyset}$.*

*$C$ is a constant value function.*

**Remark F.21.** *The proposition shows that $\mathcal{H}'_1$, as a subspace of $\mathcal{H}_1$, evolves monotonically and in opposite direction to $\mathcal{H}_2$. If we perceive the representation $\Phi$ as a filter, gaining more information from covariates facilitates multicalibration w.r.t. $\mathcal{H}_1$ (and accuracy) but hampers multicalibration w.r.t. $\mathcal{H}_2$ (and invariance). With $\mathcal{H}'_1$ and $\mathcal{H}_2$ combined together, a multicalibrated predictor is searching for an appropriate level of information filter to balance the tradeoff between accuracy and invariance.*

*Proof.* We first prove $\mathcal{H}'_{1,\Phi} \subset \mathcal{H}_{1,\Phi}$. According to Equation 122,

$$\mathcal{H}_{1,\Phi} := \{h \in C(\mathcal{X} \times \mathcal{Y}) : \mathrm{Cov}\,[h(\Phi, Y), Y | f_\Phi = v] = 0 \text{ for almost every } v \in [0,1]\}. \tag{165}$$

Since $h(\Phi) \perp Y \mid \Phi$, we have $\mathrm{Cov}\,[h(\Phi), Y|\Phi] = 0$.

$$\mathrm{Cov}\,[h(\Phi), Y|f_\Phi] \tag{166}$$
$$= \mathbb{E}\,[h(\Phi)Y|f_\Phi] - \mathbb{E}\,[h(\Phi)|f_\Phi]\,\mathbb{E}\,[Y|f_\Phi] \tag{167}$$
$$= \mathbb{E}\,[h(\Phi)Y|f_\Phi] - \mathbb{E}\,[h(\Phi)|f_\Phi]\,f_\Phi \tag{168}$$
$$= \mathbb{E}\,[\mathbb{E}\,[h(\Phi)Y|\Phi] - \mathbb{E}\,[h(\Phi)|\Phi]\,f_\Phi|f_\Phi] \tag{169}$$
$$= \mathbb{E}\,[\mathbb{E}\,[h(\Phi)Y|\Phi] - \mathbb{E}\,[h(\Phi)|\Phi]\,\mathbb{E}\,[Y|\Phi]\,|f_\Phi] \tag{170}$$
$$= \mathbb{E}\,[\mathrm{Cov}\,[h(\Phi), Y|\Phi]\,|f_\Phi] \tag{171}$$
$$= 0. \tag{172}$$

Thus, $\mathcal{H}'_{1,\Phi} \subset \mathcal{H}_{1,\Phi}$. Next, we prove the two arguments in the proposition.

1. For any $h(\Lambda) \in \mathcal{H}'_{1,\Lambda}$, since $\Phi = (\Lambda, \Omega)$, $h(\Lambda)$ is also a function of $\Phi$. Thus, we have $h(\Lambda) \in \mathcal{H}'_{1,\Phi}$. It follows that $\mathcal{H}'_{1,\Phi} \supset \mathcal{H}'_{1,\Lambda}$. Similarly we have $\mathcal{H}'_{1,X} \supset \mathcal{H}'_{1,\Phi}$ and $\mathcal{H}'_{1,\Lambda} \supset \mathcal{H}'_{1,\emptyset}$.

2. For any $h(\Lambda, \Omega, \Psi, Y) \in \mathcal{H}_{2,\Phi}$ such that $\mathbb{E}[h(\Phi, \Psi, Y)|\Phi, Y] = C_h$ for any values of $\Phi, Y$, we have

$$\mathbb{E}[h(\Lambda, \Omega, \Psi, Y)|\Lambda, Y] = \mathbb{E}\,[\mathbb{E}[h(\Lambda, \Omega, \Psi, Y)|\Lambda, \Omega, Y]\big|\Lambda, Y] \tag{173}$$
$$= \mathbb{E}\,[\mathbb{E}[h(\Phi, \Psi, Y)|\Phi, Y]\big|\Lambda, Y] \tag{174}$$
$$= \mathbb{E}[C_h|\Lambda, Y] \tag{175}$$
$$= C_h \quad \text{for any values of } \Lambda, Y. \tag{176}$$

Thus, $h(\Lambda, \Omega, \Psi, Y) \in \mathcal{H}_{2,\Lambda}$. It follows that $H_{2,\Phi} \subset \mathcal{H}_{2,\Lambda}$. Similarly, we have $\mathcal{H}_{2,X} \subset H_{2,\Phi}$ and $\mathcal{H}_{2,\Lambda} \subset \mathcal{H}_{2,\emptyset}$. Particularly for $h(x,y) \in \mathcal{H}_{2,X}$, we have $h(X, Y) = \mathbb{E}[h(X, Y)|X, Y]$ is a constant for any values of $X, Y$. $\square$

**Lemma F.22** (Globus-Harris et al. [13])**.** *Let $\mathcal{H} \subset \mathcal{X}^{\mathbb{R}}$ be such a grouping function class that $h \in \mathcal{H}$ implies $\gamma h + \rho \in \mathcal{H}$ for any $h \in \mathcal{H}$ and $\gamma, \rho \in \mathbb{R}$. If $\mathcal{H}$ satisfies the $(0,0)$-weak learning condition in Assumption F.1, a predictor $f$ is multicalibrated w.r.t. $\mathcal{H}$ if and only if $f(x) = \mathbb{E}[Y|x]$ almost surely.*

**Proposition F.23.** *For a measurable function $\Phi : \mathbb{R}^d \to \mathbb{R}^{d_\Phi}, d_\Phi \in Z^+$, a predictor $f : supp(\Phi) \to [0,1]$ is multicalibrated w.r.t. $\mathcal{H}_{1,\Phi}$ if and only if $f$ is multicalibrated w.r.t. $\mathcal{H}'_{1,\Phi}$.*

*Proof.* Since $\mathcal{H}'_{1,\Phi} \subset \mathcal{H}_{1,\Phi}$, $f$'s multicalibration w.r.t. $\mathcal{H}_{1,\Phi}$ implies $f$'s multicalibration w.r.t. $\mathcal{H}'_{1,\Phi}$.

On the other hand, $\mathcal{H}'_{1,\Phi}$ satisfies the $(0,0)$-weak learning condition with the pushforward measure on $\Phi$, because $\mathbb{E}[Y|\Phi] \in \mathcal{H}'_{1,\Phi}$. It follows from Lemma F.22 that $f$ is multicalibrated w.r.t. $\mathcal{H}'_{1,\Phi}$ implies $f(\Phi) = \mathbb{E}[Y|\Phi]$ almost surely. By the definition of $\mathcal{H}_{1,\Phi}$, $f(\Phi)$ is multicalibrated w.r.t. $\mathcal{H}_{1,\Phi}$. $\square$

## F.4 MC-PseudoLabel: An Algorithm for Extended Multicalibration

**Lemma F.24.** *Fix a model $f : \mathcal{X} \to [0,1]$. Suppose for some $v \in Range(f)$ there is an $h \in \mathcal{H}$ such that:*

$$\mathbb{E}[h(x,y)(y-v)|f(x) = v] > \alpha$$

*Let $h' = v + \eta h(x,y)$ for $\eta = \frac{\alpha}{\mathbb{E}[h(x,y)^2|f(x)=v]}$.*

*Then:*

$$\mathbb{E}[(f(x) - y)^2 - (h'(x,y) - y)^2|f(x) = v] > \frac{\alpha^2}{\mathbb{E}[h(x,y)^2|f(x) = v]}.$$

*Proof. Following [13], we have*

$$
\begin{aligned}
\mathbb{E}[(f(x) &- y)^2 - (h'(x,y) - y)^2|f(x) = v] \\
&= \mathbb{E}[(v - y)^2 - (v + \eta h(x,y) - y)^2|f(x) = v] \\
&= \mathbb{E}[v^2 - 2vy + y^2 - (v + \eta h(x,y))^2 + 2y(v + \eta h(x,y)) - y^2|f(x) = v] \\
&= \mathbb{E}[2\eta y h(x,y) - 2\eta v h(x,y) - \eta^2 h(x,y)^2|f(x) = v] \\
&= \mathbb{E}[2\eta h(x,y)(y - v) - \eta^2 h(x,y)^2|f(x) = v] \\
&> 2\eta\alpha - \eta^2 \mathbb{E}[h(x,y)^2|f(x) = v] \\
&= \frac{\alpha^2}{\mathbb{E}[h(x,y)^2|f(x) = v]}.
\end{aligned}
$$

**Theorem F.25** (Restatement of Theorem 5.1)**.** *In Algorithm 1, for $\alpha, B > 0$, if the following is satisfied:*

$$err_{t-1} - \tilde{err}_t \leq \frac{\alpha}{B}, \tag{177}$$

*the output $f'_{t-1}(x)$ is $\alpha$-approximately $\ell_2$ multicalibrated w.r.t. $\mathcal{H}_B = \{h \in \mathcal{H} : \sup h(x,y)^2 \leq B\}$.*

*Proof.* We prove by contradiction. Assume that $f_{t-1}$ is not $\alpha$-approximately multicalibrated with respect to $\mathcal{H}_B$. Then there exists $h \in \mathcal{H}_B$ such that:

$$\sum_{v \in [1/m]} P(f_{t-1}(x) = v) \left( \mathbb{E}\left[ h(x,y)(y - v) \middle| f_{t-1}(x) = v \right] \right)^2 > \alpha.$$

For each $v \in [1/m]$ define

$$\alpha_v := P(f_{t-1}(x) = v) \left( \mathbb{E}\left[ h(x,y)(y - v) \middle| f_{t-1}(x) = v \right] \right)^2.$$

Then we have $\sum_{v \in [1/m]} \alpha_v > \alpha$.

According to Lemma F.24, for each $v \in [1/m]$, there exists $h_v \in \mathcal{H}$ such that:

$$\mathbb{E}\left[(f_{t-1}(x) - y)^2 - (h_v(x,y) - y)^2 | f_{t-1}(x) = v\right] \tag{178}$$

$$> \frac{\alpha_v}{\mathbb{E}[h(x)^2 | f_{t-1}(x) = v] \cdot P(f_{t-1}(x) = v)} \tag{179}$$

$$\geq \frac{\alpha_v}{B \cdot P(f_{t-1}(x) = v)}. \tag{180}$$

Then,

$$\mathbb{E}\left[(f_{t-1}(x) - y)^2 - (\tilde{f}_t(x) - y)^2\right]$$

$$= \sum_{v \in [1/m]} P(f_{t-1}(x) = v)\mathbb{E}\left[(f_{t-1}(x) - y)^2 - (\tilde{f}_t(x) - y)^2 | f_{t-1}(x) = v\right]$$

$$= \sum_{v \in [1/m]} P(f_{t-1}(x) = v)\mathbb{E}\left[(f_{t-1}(x) - y)^2 - (h_v^t(x,y) - y)^2 | f_{t-1}(x) = v\right]$$

$$> \frac{\alpha}{B},$$

which contradicts the condition in Equation 177. $\qquad\square$

The following proposition is a direct corollary from Globus-Harris et al. [13]'s Theorem 4.3.

**Proposition F.26.** *For any distribution $D$ supported on $\mathcal{X} \times \mathcal{Y}$ and $\Phi \in \sigma(X)$, take the grouping function class $\mathcal{H} \subset \mathcal{H}'_{1,\Phi} := \{h(\Phi(x)) \in C(\mathcal{X} \times \mathcal{Y})\}$ and the predictor class $\mathcal{F} = \mathbb{R}^{\mathcal{X}}$. For any $0 < \alpha < 1, B > 0$ and an initial predictor $f_0 : \mathcal{X} \to [0,1]$ with $|Range(f_0)| \geq \frac{2B}{\alpha}$, then MC-Pseudolabel$(D, \mathcal{H}, \mathcal{F})$ halts after at most $T \leq \frac{2B}{\alpha}$ steps and outputs a model $f'_{T-1}(x)$ that is $\alpha$-approximately $\ell_2$ multicalibrated w.r.t $D$ and $\mathcal{H}_B = \{h \in \mathcal{H} : \sup h(x,y)^2 \leq B\}$.*

**Theorem F.27** (Restatement of Theorem 5.2). *Consider $X \in \mathbb{R}^d$ with $X = (\Phi, \Psi)^T$. Assume that $(\Phi, \Psi, Y)$ follows a multivariate normal distribution $\mathcal{N}_{d+1}(\mu, \Sigma)$ where the random variables are in general position such that $\Sigma$ is positive definite. We partition $\Sigma$ into blocks:*

$$\Sigma = \begin{pmatrix} \Sigma_{\Phi\Phi} & \Sigma_{\Phi\Psi} & \Sigma_{\Phi y} \\ \Sigma_{\Psi\Phi} & \Sigma_{\Psi\Psi} & \Sigma_{\Psi y} \\ \Sigma_{y\Phi} & \Sigma_{y\Psi} & \Sigma_{yy} \end{pmatrix}. \tag{181}$$

*For any distribution $D$ supported on $\mathcal{X} \times \mathcal{Y}$, take the the predictor class $\mathcal{F} = \mathbb{R}^{\mathcal{X}}$ and the grouping function class $\mathcal{H}$ as a subset of $\mathcal{H}_{2,\Phi}$, which is defined in Equation 124:*

$$\mathcal{H} = \{h : h \in \mathcal{H}_{2,\Phi} \text{ and } h(x,y) = c_x^T x + c_y y + c_b, c_x \in \mathbb{R}^d, c_y, c_b \in \mathbb{R}\}. \tag{182}$$

*For an initial predictor $f^{(0)}(x) = \mathbb{E}[Y|x]$, run MC-Pseudolabel$(D, \mathcal{H}, \mathcal{F})$ without rounding, then there exists some constant $C_x$ depending on $x$ and some constant $M(\Sigma)$ depending on $\Sigma$, such that $\left|f^{(t)}(x) - \mathbb{E}[Y|\Phi(x)]\right| \leq C_x M(\Sigma)^t$, where*

$$M(\Sigma) = (\Sigma_{yy} - \Sigma_{y\Phi}\Sigma_{\Phi\Phi}^{-1}\Sigma_{\Phi y})^{-1}(\Sigma_{y\Psi} - \Sigma_{y\Phi}\Sigma_{\Phi\Phi}^{-1}\Sigma_{\Phi\Psi}) \tag{183}$$

$$(\Sigma_{\Psi\Psi} - \Sigma_{\Psi\Phi}\Sigma_{\Phi\Phi}^{-1}\Sigma_{\Phi\Psi})^{-1}(\Sigma_{\Psi y} - \Sigma_{\Psi\Phi}\Sigma_{\Phi\Phi}^{-1}\Sigma_{\Phi y}). \tag{184}$$

*We have $0 \leq M(\Sigma) < 1$.*

**Remark F.28.** *$f^{(t)}$ and $\mathbb{E}[Y|\Phi(x)]$ are both linear. Thus, the convergence of the functions $f^{(t)}$ is equivalent as the convergence of their coefficients. The theorem essentially states that the coefficients of $f^{(t)}$ converges to those of $\mathbb{E}[Y|\Phi(x)]$ at a rate of $\mathcal{O}(M(\Sigma)^t)$.*

**Remark F.29.** *$\mathbb{E}[Y|\Phi(x)]$ is multicalibrated with respect to $\mathcal{H}$. Furthermore, any calibrated predictor on $\Phi$, denoted by $g(\Phi)$, is multicalibrated with respect to $\mathcal{H}$. This is because:*

$$\mathbb{E}[h(X,Y)(Y - g(\Phi))|g(\Phi)] = \mathbb{E}\left[\mathbb{E}\left[h(X,Y)(Y - g(\Phi))|\Phi, Y\right]|g(\Phi)\right] \tag{185}$$

$$= \mathbb{E}\left[C_h(Y - g(\Phi))|g(\Phi)\right] \tag{186}$$

$$= 0. \tag{187}$$

*However, $\mathbb{E}[Y|\Phi]$ is the most accurate predictor among all multicalibrated $g(\Phi)$.*

**Remark F.30.** *The convergence rate $M(\Sigma)$ does not depend on the dimension $d$ of covariates. When $Y \perp \Psi \mid \Phi$ implying that $\Phi$ is sufficient for prediction, following from $\mathbb{E}[Y|\Phi, \Psi] = \mathbb{E}[Y|\Phi]$:*

$$\Sigma_{\Psi y} = \Sigma_{\Psi \Phi} \Sigma_{\Phi\Phi}^{-1} \Sigma_{\Phi y}. \tag{188}$$

*It follows that $M(\Sigma) = 0$ and the algorithm will converge in one step.*

*On the other hand, when $Y$ and $\Psi$ are linearly dependent given $\Phi$ such that $\Sigma$ is singular, which violates positive definiteness, following from the proof below:*

$$\Sigma_{yy} - \Sigma_{y\Phi} \Sigma_{\Phi\Phi}^{-1} \Sigma_{\Phi y} = (\Sigma_{y\Psi} - \Sigma_{y\Phi} \Sigma_{\Phi\Phi}^{-1} \Sigma_{\Phi\Psi})(\Sigma_{\Psi\Psi} - \Sigma_{\Psi\Phi} \Sigma_{\Phi\Phi}^{-1} \Sigma_{\Phi\Psi})^{-1}(\Sigma_{\Psi y} - \Sigma_{\Psi\Phi} \Sigma_{\Phi\Phi}^{-1} \Sigma_{\Phi y}). \tag{189}$$

*It follows that $M(\Sigma) = 1$ and the algorithm can't converge.*

*So the convergence rate depends on the singularity of the problem. Since the algorithm converges to a predictor that does not depend on $\Psi$, stronger the "spurious" correlation between $Y$ and $\Psi$ given $\Phi$ in the distribution D, the algorithm takes longer to converge.*

*Proof.* Without loss of generality, assume $\mu = 0$.

Let $c = (c_x, c_y, c_b)^T = (c_\Phi, c_\Psi, c_y, c_b)^T$. Denote dimensions of $\Phi, \Psi$ by $d_\Phi, d_\Psi$.

Let $\mathbb{E}[Y|\Phi, \Psi] = (\alpha_\Phi^\star)^T \Phi + (\alpha_\Psi^\star)^T \Psi$ and $\mathbb{E}[\Psi|\Phi, Y] = (\beta_\Phi^\star)^T \Phi + (\beta_y^\star)^T Y$ with $\alpha_\Phi^\star \in \mathbb{R}^{d_\Phi}$, $\alpha_\Psi^\star \in \mathbb{R}^{d_\Psi}, \beta_\Phi^\star \in \mathbb{R}^{d_\Phi \times d_\Psi}, \beta_y^\star \in \mathbb{R}^{1 \times d_\Psi}$.

We have:

$$\begin{pmatrix} \alpha_\Phi^\star \\ \alpha_\Psi^\star \end{pmatrix} = \begin{pmatrix} \Sigma_{\Phi\Phi} & \Sigma_{\Phi\Psi} \\ \Sigma_{\Psi\Phi} & \Sigma_{\Psi\Psi} \end{pmatrix}^{-1} \begin{pmatrix} \Sigma_{\Phi y} \\ \Sigma_{\Psi y} \end{pmatrix}. \tag{190}$$

$$\begin{pmatrix} \beta_\Phi^\star \\ \beta_y^\star \end{pmatrix} = \begin{pmatrix} \Sigma_{\Phi\Phi} & \Sigma_{\Phi y} \\ \Sigma_{y\Phi} & \Sigma_{yy} \end{pmatrix}^{-1} \begin{pmatrix} \Sigma_{\Phi\Psi} \\ \Sigma_{y\Psi} \end{pmatrix}. \tag{191}$$

According to Theorem F.15, $\mathbb{E}[h(X, Y)|\Phi, Y]$ is a constant for different values of $\Phi, Y$.

$$\mathbb{E}[h(X, Y)|\Phi, Y] = c_\Phi^T \Phi + c_y^T Y + c_\Psi^T \mathbb{E}[\Psi|\Phi, Y] + c_b \tag{192}$$

$$= c_\Phi^T \Phi + c_y^T Y + c_\Psi^T \begin{pmatrix} \Sigma_{\Psi\Phi} & \Sigma_{\Psi y} \end{pmatrix} \begin{pmatrix} \Sigma_{\Phi\Phi} & \Sigma_{\Phi y} \\ \Sigma_{y\Phi} & \Sigma_{yy} \end{pmatrix}^{-1} \begin{pmatrix} \Phi \\ Y \end{pmatrix} + c_b. \tag{193}$$

This implies:

$$\begin{pmatrix} \Sigma_{\Phi\Phi} & \Sigma_{\Phi y} \\ \Sigma_{y\Phi} & \Sigma_{yy} \end{pmatrix}^{-1} \begin{pmatrix} \Sigma_{\Phi\Psi} \\ \Sigma_{y\Psi} \end{pmatrix} c_\Psi + \begin{pmatrix} c_\Phi \\ c_y \end{pmatrix} = 0. \tag{194}$$

Rearranging to:

$$\begin{pmatrix} \Sigma_{\Phi\Phi} & \Sigma_{\Phi\Psi} & \Sigma_{\Phi y} \\ \Sigma_{y\Phi} & \Sigma_{y\Psi} & \Sigma_{yy} \end{pmatrix} c = 0. \tag{195}$$

Let $f^{(t)} = (\alpha_\Phi^{(t)})^T \Phi + (\alpha_\Psi^{(t)})^T \Psi$ and $\tilde{f}^{(t)} = (\tilde\alpha_\Phi^{(t)})^T \Phi + (\tilde\alpha_\Psi^{(t)})^T \Psi + (\tilde\alpha_y^{(t)})^T Y$. We claim that $\tilde\alpha_\Psi^{(t)} = 0$. Otherwise, consider $\tilde{f}^{(t)'} = (\tilde\alpha_\Phi^{(t)})^T \Phi + (\tilde\alpha_\Psi^{(t)})^T \mathbb{E}[\Psi|\Phi, Y] + (\tilde\alpha_y^{(t)})^T Y$.

$$\mathbb{E}[\tilde{f}^{(t)} - \tilde{f}^{(t)'}|\Psi, Y] = \mathbb{E}[(\tilde\alpha_\Psi^{(t)})^T \Psi|\Phi, Y] - (\tilde\alpha_\Psi^{(t)})^T \mathbb{E}[\Psi|\Phi, Y] = 0. \tag{196}$$

Thus, $\tilde{f}^{(t)} - \tilde{f}^{(t)'} \in \mathcal{H}$.

On the other hand,

$$\mathbb{E}[(Y - \tilde{f}^{(t)})^2] - \mathbb{E}[(Y - \tilde{f}^{(t)'})^2] \tag{197}$$

$$= \mathbb{E}\left[(\tilde\alpha_\Psi^{(t)})^T \left[\mathbb{E}[\Psi\Psi^T|\Phi, Y] - \mathbb{E}[\Psi|\Phi, Y]\mathbb{E}[\Psi^T|\Phi, Y]\right] \tilde\alpha_\Psi^{(t)}\right] \tag{198}$$

$$> 0. \tag{199}$$

The inequality follows from the fact that $\mathbb{E}[\Psi\Psi^T|\Phi, Y] - \mathbb{E}[\Psi|\Phi, Y]\mathbb{E}[\Psi^T|\Phi, Y]$ is the covariance matrix of $\Psi|\Phi, Y$, which is positive definite because $\Sigma$ is positive definite. The inequality contradicts with the definition of $\tilde{f}^{(t)}$. Thus, $\tilde{f}^{(t)} = (\tilde{\alpha}_\Phi^{(t)})^T\Phi + (\tilde{\alpha}_y^{(t)})^TY$.

Define a matrix $C \in \mathbb{R}^{(d+1)\times d_t}$ whose columns support the solution space of Equation 195. Then $H := C^T(S, T, Y)^T \in \mathbb{R}^{d_t}$ is a random vector. According to the definition of $\tilde{f}^{(t)}$,

$$\tilde{f}^{(t+1)} = \mathbb{E}[Y|f^{(t)}, H] \tag{200}$$

$$= k^{(t)}f^{(t)} + (c_\Phi^{(t)})^T\Phi + (c_\Psi^{(t)})^T\Psi + (c_y^{(t)})^TY \tag{201}$$

$$= k^{(t)}(\alpha_\Phi^{(t)})^T\Phi + k^{(t)}(\alpha_\Psi^{(t)})^T\Psi + (c_\Phi^{(t)})^T\Phi + (c_\Psi^{(t)})^T\Psi + (c_y^{(t)})^TY. \tag{202}$$

In the above equation, $k^{(t)} \in \mathbb{R}$.

Since $\tilde{\alpha}_\Psi^{(t+1)} = k^{(t)}\alpha_\Psi^{(t)} + c_\Psi^{(t)} = 0$, we have $c_\Psi^{(t)} = -k^{(t)}\alpha_\Psi^{(t)}$. Substituting into Equation 194:

$$\begin{pmatrix} c_\Phi^{(t)} \\ c_y^{(t)} \end{pmatrix} = k^{(t)} \begin{pmatrix} \Sigma_{\Phi\Phi} & \Sigma_{\Phi y} \\ \Sigma_{y\Phi} & \Sigma_{yy} \end{pmatrix}^{-1} \begin{pmatrix} \Sigma_{\Phi\Psi} \\ \Sigma_{y\Psi} \end{pmatrix} \alpha_\Psi^{(t)}. \tag{203}$$

Substituting into Equation 202:

$$\begin{pmatrix} \tilde{\alpha}_\Phi^{(t+1)} \\ \tilde{\alpha}_y^{(t+1)} \end{pmatrix} = k^{(t)} \begin{pmatrix} \tilde{\alpha}_\Phi^{(t)} \\ 0 \end{pmatrix} + \begin{pmatrix} c_\Phi^{(t)} \\ c_y^{(t)} \end{pmatrix} \tag{204}$$

$$= k^{(t)} \begin{pmatrix} I_{d_\Phi} & \beta_\Phi^\star \\ 0 & \beta_y^\star \end{pmatrix} \begin{pmatrix} \alpha_\Phi^{(t)} \\ \alpha_\Psi^{(t)} \end{pmatrix}. \tag{205}$$

Then we have $f^{(t+1)} = \mathbb{E}[\tilde{f}^{(t+1)}|\Phi, \Psi] = (\tilde{\alpha}_\Phi^{(t+1)})^T\Phi + (\tilde{\alpha}_y^{(t+1)})^T\mathbb{E}[Y|\Phi, \Psi]$.

This is equivalent as:

$$\begin{pmatrix} \alpha_\Phi^{(t+1)} \\ \alpha_\Psi^{(t+1)} \end{pmatrix} = \begin{pmatrix} \tilde{\alpha}_\Phi^{(t+1)} \\ 0 \end{pmatrix} + \begin{pmatrix} \Sigma_{\Phi\Phi} & \Sigma_{\Phi\Psi} \\ \Sigma_{\Psi\Phi} & \Sigma_{\Psi\Psi} \end{pmatrix}^{-1} \begin{pmatrix} \Sigma_{\Phi y} \\ \Sigma_{\Psi y} \end{pmatrix} \tilde{\alpha}_y^{(t+1)} \tag{206}$$

$$= \begin{pmatrix} I_{d_\Phi} & \alpha_\Phi^\star \\ 0 & \alpha_\Psi^\star \end{pmatrix} \begin{pmatrix} \tilde{\alpha}_\Phi^{(t+1)} \\ \tilde{\alpha}_y^{(t+1)} \end{pmatrix}. \tag{207}$$

Combining the two equations above, we have:

$$\begin{pmatrix} \alpha_\Phi^{(t+1)} \\ \alpha_\Psi^{(t+1)} \end{pmatrix} = k^{(t)} \begin{pmatrix} I_{d_\Phi} & \alpha_\Phi^\star \\ 0 & \alpha_\Psi^\star \end{pmatrix} \begin{pmatrix} I_{d_\Phi} & \beta_\Phi^\star \\ 0 & \beta_y^\star \end{pmatrix} \begin{pmatrix} \alpha_\Phi^{(t)} \\ \alpha_\Psi^{(t)} \end{pmatrix} \tag{208}$$

$$= k^{(t)} \begin{pmatrix} I_{d_\Phi} & \beta_\Phi^\star + \alpha_\Phi^\star\beta_y^\star \\ 0 & \alpha_\Psi^\star\beta_y^\star \end{pmatrix} \begin{pmatrix} \alpha_\Phi^{(t)} \\ \alpha_\Psi^{(t)} \end{pmatrix}. \tag{209}$$

Thus,

$$\begin{pmatrix} \alpha_\Phi^{(t)} \\ \alpha_\Psi^{(t)} \end{pmatrix} = K^{(t)} \begin{pmatrix} I_{d_\Phi} & \beta_\Phi^\star + \alpha_\Phi^\star\beta_y^\star \\ 0 & \alpha_\Psi^\star\beta_y^\star \end{pmatrix}^t \begin{pmatrix} \alpha_\Phi^{(0)} \\ \alpha_\Psi^{(0)} \end{pmatrix} \tag{210}$$

$$= K^{(t)} \begin{pmatrix} I_{d_\Phi} & \beta_\Phi^\star + \alpha_\Phi^\star\beta_y^\star \\ 0 & \alpha_\Psi^\star\beta_y^\star \end{pmatrix}^t \begin{pmatrix} \alpha_\Phi^\star \\ \alpha_\Psi^\star \end{pmatrix}. \tag{211}$$

In the above equation, $K^{(t)} = \prod_{0 \leq u < t} k^{(u)}$.

Define $\hat{f}^{(t)} = (\hat{\alpha}_\Phi^{(t)})^T\Phi + (\hat{\alpha}_\Psi^{(t)})^T\Psi$, where

$$\begin{pmatrix} \hat{\alpha}_\Phi^{(t)} \\ \hat{\alpha}_\Psi^{(t)} \end{pmatrix} = \begin{pmatrix} I_{d_\Phi} & \beta_\Phi^\star + \alpha_\Phi^\star\beta_y^\star \\ 0 & \alpha_\Psi^\star\beta_y^\star \end{pmatrix}^t \begin{pmatrix} \alpha_\Phi^\star \\ \alpha_\Psi^\star \end{pmatrix}. \tag{212}$$

In the following we show $\hat{\alpha}_{\Psi}^{(t)} \to 0$. Since $\hat{\alpha}_{\Psi}^{(t)} = (\alpha_{\Psi}^{\star}\beta_y^{\star})^t \alpha_{\Psi}^{\star}$, and $\alpha_{\Psi}^{\star}\beta_y^{\star} \in \mathbb{R}^{d_{\Psi} \times d_{\Psi}}$ has exactly one nonzero eigenvalue $\beta_y^{\star}\alpha_{\Psi}^{\star}$, we just have to show $|\beta_y^{\star}\alpha_{\Psi}^{\star}| < 1$.

$$\beta_y^{\star}\alpha_{\Psi}^{\star} = (\Sigma_{yy} - \Sigma_{y\Phi}\Sigma_{\Phi\Phi}^{-1}\Sigma_{\Phi y})^{-1} \tag{213}$$

$$(\Sigma_{y\Psi} - \Sigma_{y\Phi}\Sigma_{\Phi\Phi}^{-1}\Sigma_{\Phi\Psi})(\Sigma_{\Psi\Psi} - \Sigma_{\Psi\Phi}\Sigma_{\Phi\Phi}^{-1}\Sigma_{\Phi\Psi})^{-1}(\Sigma_{\Psi y} - \Sigma_{\Psi\Phi}\Sigma_{\Phi\Phi}^{-1}\Sigma_{\Phi y}). \tag{214}$$

Since, $(\Sigma_{yy} - \Sigma_{y\Phi}\Sigma_{\Phi\Phi}^{-1}\Sigma_{\Phi y})$ and $(\Sigma_{\Psi\Psi} - \Sigma_{\Psi\Phi}\Sigma_{\Phi\Phi}^{-1}\Sigma_{\Phi\Psi})$ are both Schur complements of $\Sigma$'s principal submatrix, they are still positive definite.

Thus,

$$\Sigma_{yy} - \Sigma_{y\Phi}\Sigma_{\Phi\Phi}^{-1}\Sigma_{\Phi y} > 0. \tag{215}$$

$$(\Sigma_{y\Psi} - \Sigma_{y\Phi}\Sigma_{\Phi\Phi}^{-1}\Sigma_{\Phi\Psi})(\Sigma_{\Psi\Psi} - \Sigma_{\Psi\Phi}\Sigma_{\Phi\Phi}^{-1}\Sigma_{\Phi\Psi})^{-1}(\Sigma_{\Psi y} - \Sigma_{\Psi\Phi}\Sigma_{\Phi\Phi}^{-1}\Sigma_{\Phi y}) \geq 0. \tag{216}$$

It follows that $\beta_y^{\star}\alpha_{\Psi}^{\star} \geq 0$. So we just have to show $\beta_y^{\star}\alpha_{\Psi}^{\star} < 1$.

Since $\det(\Sigma) > 0$, by applying row addition on $\Sigma$, we have:

$$\det \begin{pmatrix} \Sigma_{\Phi\Phi} & \Sigma_{\Phi\Psi} & \Sigma_{\Phi y} \\ 0 & \Sigma_{\Psi\Psi} - \Sigma_{\Psi\Phi}\Sigma_{\Phi\Phi}^{-1}\Sigma_{\Phi\Psi} & \Sigma_{\Psi y} - \Sigma_{\Psi\Phi}\Sigma_{\Phi\Phi}^{-1}\Sigma_{\Phi y} \\ 0 & \Sigma_{y\Psi} - \Sigma_{y\Phi}\Sigma_{\Phi\Phi}^{-1}\Sigma_{\Phi\Psi} & \Sigma_{yy} - \Sigma_{y\Phi}\Sigma_{\Phi\Phi}^{-1}\Sigma_{\Phi y} \end{pmatrix} > 0. \tag{217}$$

It follows that:

$$(\Sigma_{yy} - \Sigma_{y\Phi}\Sigma_{\Phi\Phi}^{-1}\Sigma_{\Phi y}) \tag{218}$$

$$- (\Sigma_{y\Psi} - \Sigma_{y\Phi}\Sigma_{\Phi\Phi}^{-1}\Sigma_{\Phi\Psi})(\Sigma_{\Psi\Psi} - \Sigma_{\Psi\Phi}\Sigma_{\Phi\Phi}^{-1}\Sigma_{\Phi\Psi})^{-1}(\Sigma_{\Psi y} - \Sigma_{\Psi\Phi}\Sigma_{\Phi\Phi}^{-1}\Sigma_{\Phi y}) > 0. \tag{219}$$

Rearranging to:

$$\beta_y^{\star}\alpha_{\Psi}^{\star} = (\Sigma_{yy} - \Sigma_{y\Phi}\Sigma_{\Phi\Phi}^{-1}\Sigma_{\Phi y})^{-1} \tag{220}$$

$$(\Sigma_{y\Psi} - \Sigma_{y\Phi}\Sigma_{\Phi\Phi}^{-1}\Sigma_{\Phi\Psi})(\Sigma_{\Psi\Psi} - \Sigma_{\Psi\Phi}\Sigma_{\Phi\Phi}^{-1}\Sigma_{\Phi\Psi})^{-1}(\Sigma_{\Psi y} - \Sigma_{\Psi\Phi}\Sigma_{\Phi\Phi}^{-1}\Sigma_{\Phi y}) \tag{221}$$

$$< 1. \tag{222}$$

Thus, $0 \leq \beta_y^{\star}\alpha_{\Psi}^{\star} < 1$.

In the following, we show $\hat{\alpha}_{\Phi}^{(t)} \to \Sigma_{\Phi\Phi}^{-1}\Sigma_{\Phi y}$. By Equation 212 and $|\beta_y^{\star}\alpha_{\Psi}^{\star}| < 1$, we have:

$$\hat{\alpha}_{\Phi}^{(t)} = \alpha_{\Phi}^{\star} + (\beta_{\Phi}^{\star} + \alpha_{\Phi}^{\star}\beta_y^{\star}) \sum_{0 \leq u < t} (\alpha_{\Psi}^{\star}\beta_y^{\star})^u \alpha_{\Psi}^{\star} \tag{223}$$

$$\xrightarrow{t \to \infty} \alpha_{\Phi}^{\star} + (\beta_{\Phi}^{\star} + \alpha_{\Phi}^{\star}\beta_y^{\star})(I_{d_{\Psi}} - \alpha_{\Psi}^{\star}\beta_y^{\star})^{-1}\alpha_{\Psi}^{\star} \tag{224}$$

$$= \alpha_{\Phi}^{\star} + (\beta_{\Phi}^{\star} + \alpha_{\Phi}^{\star}\beta_y^{\star})(1 - \beta_y^{\star}\alpha_{\Psi}^{\star})^{-1}\alpha_{\Psi}^{\star} \tag{225}$$

$$= \frac{\alpha_{\Phi}^{\star} + \beta_{\Phi}^{\star}\alpha_{\Psi}^{\star}}{1 - \beta_y^{\star}\alpha_{\Psi}^{\star}}. \tag{226}$$

Equation 225 follows from the fact that $(I_{d_{\Psi}} - \alpha_{\Psi}^{\star}\beta_y^{\star})\alpha_{\Psi}^{\star} = (1 - \beta_y^{\star}\alpha_{\Psi}^{\star})\alpha_{\Psi}^{\star}$.

Define $\gamma_{\Phi}^{\star} = \Sigma_{\Phi\Phi}^{-1}\Sigma_{\Phi y}$ such that $\mathbb{E}[Y|\Phi] = (\gamma_{\Phi}^{\star})^T\Phi$. We have

$$\mathbb{E}\left[\mathbb{E}\left[\mathbb{E}\left[Y|\Phi, \Psi\right]|\Phi, Y\right]|\Phi\right] = \mathbb{E}[Y|\Phi]. \tag{227}$$

$$\Leftrightarrow \mathbb{E}\left[\mathbb{E}\left[(\alpha_{\Phi}^{\star})^T\Phi + (\alpha_{\Psi}^{\star})^T\Psi|\Phi, Y\right]|\Phi\right] = (\gamma_{\Phi}^{\star})^T\Phi. \tag{228}$$

$$\Leftrightarrow \mathbb{E}\left[(\alpha_{\Phi}^{\star})^T\Phi + (\alpha_{\Psi}^{\star})^T(\beta_{\Phi}^{\star})^T\Phi + (\alpha_{\Psi}^{\star})^T(\beta_y^{\star})^TY|\Phi\right] = (\gamma_{\Phi}^{\star})^T\Phi. \tag{229}$$

$$\Leftrightarrow (\alpha_{\Phi}^{\star})^T\Phi + (\alpha_{\Psi}^{\star})^T(\beta_{\Phi}^{\star})^T\Phi + (\alpha_{\Psi}^{\star})^T(\beta_y^{\star})^T(\gamma_{\Phi}^{\star})^T\Phi = (\gamma_{\Phi}^{\star})^T\Phi. \tag{230}$$

$$\Leftrightarrow \alpha_{\Phi}^{\star} + \beta_{\Phi}^{\star}\alpha_{\Psi}^{\star} + \beta_y^{\star}\alpha_{\Psi}^{\star}\gamma_{\Phi}^{\star} = \gamma_{\Phi}^{\star}. \tag{231}$$

$$\Leftrightarrow \frac{\alpha_{\Phi}^{\star} + \beta_{\Phi}^{\star}\alpha_{\Psi}^{\star}}{1 - \beta_y^{\star}\alpha_{\Psi}^{\star}} = \gamma_{\Phi}^{\star}. \tag{232}$$

Thus, $\hat{\alpha}_{\Phi}^{(t)} \to \Sigma_{\Phi\Phi}^{-1}\Sigma_{\Phi y}$. Subsequently, $\hat{f}^{(t)} \to (\Sigma_{\Phi\Phi}^{-1}\Sigma_{\Phi y})^T\Phi = \mathbb{E}[Y|\Phi]$. The convergence ratio is $M(\Sigma) = \beta_y^{\star}\alpha_{\Psi}^{\star}$.

We have:

$$\begin{pmatrix} \tilde{\alpha}_\Phi^{(t+1)} \\ \tilde{\alpha}_y^{(t+1)} \end{pmatrix} = K^{(t+1)} \begin{pmatrix} I_{d_\Phi} & \beta_\Phi^\star \\ 0 & \beta_y^\star \end{pmatrix} \begin{pmatrix} \hat{\alpha}_\Phi^{(t)} \\ \hat{\alpha}_\Psi^{(t)} \end{pmatrix}, \tag{233}$$

where

$$\begin{pmatrix} I_{d_\Phi} & \beta_\Phi^\star \\ 0 & \beta_y^\star \end{pmatrix} \begin{pmatrix} \hat{\alpha}_\Phi^{(t)} \\ \hat{\alpha}_\Psi^{(t)} \end{pmatrix} \xrightarrow{t \to \infty} \begin{pmatrix} I_{d_\Phi} & \beta_\Phi^\star \\ 0 & \beta_y^\star \end{pmatrix} \begin{pmatrix} \Sigma_{\Phi\Phi}^{-1} \Sigma_{\Phi y} \\ 0 \end{pmatrix} \tag{234}$$

$$= \begin{pmatrix} \Sigma_{\Phi\Phi}^{-1} \Sigma_{\Phi y} \\ 0 \end{pmatrix}. \tag{235}$$

Define $\hat{\tilde{f}}^{(t)} = (\hat{\tilde{\alpha}}_\Phi^{(t)})^T \Phi + (\hat{\tilde{\alpha}}_y^{(t)})^T Y$, where

$$\begin{pmatrix} \hat{\tilde{\alpha}}_\Phi^{(t+1)} \\ \hat{\tilde{\alpha}}_y^{(t+1)} \end{pmatrix} = \begin{pmatrix} I_{d_\Phi} & \beta_\Phi^\star \\ 0 & \beta_y^\star \end{pmatrix} \begin{pmatrix} \hat{\alpha}_\Phi^{(t)} \\ \hat{\alpha}_\Psi^{(t)} \end{pmatrix}. \tag{236}$$

Thus, $\hat{\tilde{f}}^{(t)} \to \mathbb{E}[Y|\Phi]$. Since $\tilde{f}^{(t)} = K^{(t)} \hat{\tilde{f}}^{(t)}$, we have $\tilde{f}^{(t)} = \mathbb{E}[Y|\hat{\tilde{f}}^{(t)}] \to \mathbb{E}[Y|\Phi]$.

Subsequently, $f^{(t)} = \mathbb{E}[\tilde{f}^{(t)}|\Phi, \Psi] \to \mathbb{E}[Y|\Phi]$.

$\square$

# G  Limitations

Both our theory and algorithm focuses on the bounded regression setting. The definition of extended multicalibration does not depend on the risk function. However, the analysis of the maximal grouping function class as a linear space assumes a continuous probability distribution of observations, implying a continuous target domain. The convergence of MC-Pseudolabel is also established in a regression setting. All experiments are performed on regression tasks. As most algorithms for out-of-distribution generalization are set up with classification problems, we fill the gap for regression and leave an extension to general risk functions for future work.

