# OpenReview forum: "Bridging Multicalibration and Out-of-distribution Generalization Beyond Covariate Shift"
_NeurIPS.cc/2024/Conference — NeurIPS 2024 poster_

### Official Review · Reviewer_ezjY · 2024-06-19

**Soundness:** 3
**Presentation:** 1
**Contribution:** 3
**Rating:** 5
**Confidence:** 3

**Summary:**

The authors propose utilizing multicalibration algorithms to achieve distributional robustness w.r.t. both concept and covariate shift. They do this by allowing subgroups to not only be a function of the features X (as is standard in multigroup fairness), but also a function of the label Y. There are numerous theoretical and experimental results. The first result (thm 3.2) states that under closure properties of the hypothesis class $\mathcal{H}$ w.r.t. possible distribution shifts, and a weak-learnability assumption on $\mathcal{H}$, multicalibration w.r.t. a particular set of subgroups $\mathcal{H}_1$ guarantees that the resulting predictor will achieve good error (relative to the bayes optimal predictor) under any possible covariate shifted distribution $P_T$.

In theorem 3.1., the authors show that an L2-multicalibrated predictor f cannot be significantly improved by any post-processing function in order to achieve robustness to concept shift. The authors then show an equivalence between approximate multicalibration and a notion called approximate invariance from prior work (theorem 3.4). Invariance is a property of a representation $\Phi$ which says that there exists a predictor $g^*$ based only on $\Phi$ which achieves good error across all environments. The equivalence between the two notions is up to considering subgroup functions defined by the density ratios between the target environments and the source distribution, similar to the result in theorem 3.1.

Next, the authors demonstrate that (joint) multicalibration cannot always be achieved for all jointly defined subgroups. Thus, they are interested in the maximal feasible set of jointly defined groups for which we can achieve multicalibration towards. These groups turn out to lie in a linear space (Prop. 4.1.) and have a well-defined spanning set which looks similar to an invariant predictor (Thm. 4.2). Finally, they show that the subgroup function space can be decomposed into an invariant and multicalibrated part (Thm. 4.3).

Lastly, experiments are conducted which demonstrate the usefulness of multicalibration in achieving distributional robustness for simple neural networks on the PovertyMap and ACSIncome datasets.

**Strengths:**

Originality: To my knowledge, the work is very original, especially the consideration of the maximal grouping function space. While extending subgroups to be defined on the label as well as feature is a natural generalization, the authors have unique function space decomposition results (section 4), which are interesting in their own right and whose proofs utilize classical results from linear algebra. Substantial theory is developed through most of the paper, mostly surrounding the inclusion of particular density ratios in the subgroup function classes. I think the results taken together are very original and interesting!

Significance: understanding what is possible with families of multicalibration post-processing algorithms is a very important and interesting direction, and combining this with out of distribution generalization is an area which is quite interesting.

**Weaknesses:**

I have not checked the proofs thoroughly.

I believe the paper has much room for improvement in terms of clarity, organization, and discussion. In isolation, these may be minor, but taken together, the writing poses a major limitation to the greater community understanding the work (as well as its position in the literature). Throughout, I have asked explicit questions which I would like the authors to address, and labeled them as Q1, Q2, etc.

W1: Firstly, I believe the introduction requires substantial re-writing. Currently, it makes sense to someone who is familiar with both the multicalibration and distributional robustness literature (most people are not). For example, the first sentence of the paper mentions both out-of-distribution generalization and multicalibration, without motivating either of these communities. (Q1) Why might we want to connect these notions? What do we stand to gain from applying multicalibration, and what does multicalibration address that the (substantial) work on OOD generalization has not considered? Perhaps these questions are answered implicitly elsewhere in the work (and I missed them), but I believe that the authors should do a better job of framing the problem they are trying to solve in the context of the literature.

W2: I believe that an explicit discussion of related work in the introduction can be very helpful. I understand that the authors defer all related work discussion to the appendix, but as someone familiar with the multicalibration literature but not distributional robustness, the main paper was hard to follow.

W3: Little interpretation of results. As this is predominantly a theoretical paper, I understand that it can be challenging to concisely state and provide intuitive explanations of each result. However, I believe that the paper can be substantially improved with some additional intuition. Even understanding the theorem statements in section 4 took substantial effort from my part.

As another example, in Section 2.2: line 101-106: it seems implied that these assumptions are from previous work, but it would be good to state explicitly. (Assuming that is the case). Further, any further discussion on the assumptions would be appreciated. I believe it could be useful to state the assumptions in 2.2 as two separate assumptions, in order to make references to them in the following theorem statement more clear.
There is no discussion after the statement of the theorem 2.3. What is the class H_1, and why does it help us achieve robustness to distribution shift? How does it relate to the two assumptions stated previously?
Also, how does Kim et al. [22] relate to theorem 2.3? These should all be explicitly discussed.

As a positive example, the authors do provide some discussion in lines 175-178. Expanding and including similar discussion throughout the paper can help make the ideas more parseable to the unfamiliar reader.

W4: Section 4 is extremely difficult to parse. This also connects to W3, as there is little interpretation of the results. What are the implications of the grouping functions being a linear space? Why is the decomposition in theorem 4.3 a Minkowski sum?  Also, how do these results connect to the algorithm developed in section 5?

Minor comments
35: The authors use the term “joint” grouping functions, but have only implicitly described them in the previous paragraph. It would be clearer to concretely mention that you call these subgroups which depend on both X and Y joint grouping functions.

41-43: The definition of invariance / IRM is not very clear here. I think it is ok to defer a formal description to later, and just mention that you provide an equivalence between the invariant learning framework and multicalibration.

In line 120, there is some motivation for covariate shift and concept shift. I believe this discussion should go far earlier, perhaps in the introduction.

**Questions:**

Q1. How does section 5 relate to the rest of the work? Why do we need another multicalibration algorithm? What problem is MC-PseudoLabel solving which existing OOD / domain adaptation algorithms do not already solve? I believe that these are critical and should be stated explicitly.

Q2. Line 83: “We say that f is $\alpha$-approximately calibrated if $h \equiv 1$.” What does this mean? I think you mean if $\mathcal{H}$ includes the constant function $h\equiv 1$, then we can say that f is $\alpha$ approximately calibrated.

Q3. Line 18: “Multicalibration is a strengthening  of calibration, which requires a predictor f to be correct on average within each level set:” For clarity, perhaps you could introduce calibration first in an isolated way, without mentioning multicalibration? The discussion here could potentially be misconstrued by the reader.

Q4. Line 80: Definition 2.1. Outside of extending to joint subgroups h(X) -> h(X, Y), how does this definition relate to existing multicalibration definitions?

Q5. Theorem  3.1: In the appendix proof of this theorem, the authors state a similarity to Blasiok et al. [5] w.r.t. the theorem statement. How similar is the proof, and what complications arise from needing to consider joint grouping functions? This should be in the main paper, especially since others may notice a resemblance between the results as well.

Q6. Line 180: “Section 3 inspires one to construct richer grouping function classes for stronger generalizability”. Why is this the case, or can you expand on what this means exactly? Is the idea that due to theorem 3.4, we understand that multicalibration w.r.t. Sufficiently rich joint grouping functions gives us a nice property, but we want to efficiently achieve this which may be difficult in practice?

**Limitations:**

The authors have stated the main limitation of the work, which is that most OOD generalization papers deal with classification, not regression. However, I see this as a feature, since regression can also be a useful consideration in practice.

---

> ### Author Rebuttal · Authors · 2024-08-06
>
> Thank you for your thorough review and efforts to improve our paper! We appreciate your recognition of our contribution's originality and significance. We believe we have included relevant details in our submission that address many of your questions. We will incorporate your questions and feedback to further enhance clarity, which we address below.
>
> ### W1,Q1: Which OOD problem is solved by applying multi-calibration ?
>
> The first advantage of multi-calibration is **universal adaptability**, i.e., generalization to any target distribution with a density ratio captured by the grouping function class [22]. Existing OOD methods target a specific distribution for domain adaptation [A1] or multiple subpopulations for domain generalization [16]. Universal adaptability is central to studies connecting multi-calibration and distribution shift, though restricted to robust statistical inference under covariate shift (L23-28, L585-589). Our research considers a general uncertainty set of target distributions (L69-73) and extends universal adaptability to prediction tasks and concept shift (L39-45).
>
> Second, our algorithm offers a post-processing optimization framework more **computationally efficient** than current OOD generalization techniques (L55-64). MC-Pseudolabel, using a trained model as input, involves supervised regressions adding only linear regression overhead compared to a standard backward pass. Established OOD methods perform bi-level optimization or multi-objective learning, which involves higher-order derivatives or adversarial training.
>
> [A1] Zhao, H., Des Combes, R. T., Zhang, K., & Gordon, G. On learning invariant representations for domain adaptation.
>
> ### W1,Q1: Why do we need another multi-calibration algorithm?
>
> In section 5, we develop a new multi-calibration algorithm for joint grouping functions (dependent on x and y) because we cannot trivially extend existing algorithms for covariate-based grouping functions like LSBoost. The comparison between MC-Pseudolabel and LSBoost is detailed in L264-273. The main obstacle is that those algorithms produce predictors taking y as input. As a solution, we project the predictors back to x-based functions by tuning models with pseudo labels. This projection substantially changes optimization dynamics by disrupting the monotonicity of risks which LSBoost relies on, and also results in a different proof of convergence.
>
> ### W2: Moving the discussion of related work from appendix to introduction.
>
> We appreciate your suggestion. Complementing our current review of multi-calibration studies and their connection to distribution shift (L20-28), we will expand the discussion on key related methods for distributional robustness (L59-61) to highlight gaps in universal adaptability and computational efficiency.
>
> ### W3: Interpretation of Theorem 2.3.
>
> We expand the discussion on Theorem 2.3 in L96-104, focusing on its connection to two established results and assumptions. This theorem bridges results from Kim et al. and Globus-Harris et al. (L98) and serves as a warm-up solution to covariate shift before focusing on concept shift. Kim et al. show multi-calibrated predictors remain multi-calibrated under covariate shift, with Assumption 2.2.1. Globus-Harris et al. show multi-calibrated predictors approach Bayes optimality in a single distribution, with Assumption 2.2.2. Combining both assumptions, the theorem shows multi-calibrated predictors approach Bayes optimality in target distributions under covariate shift. ***We are ready to further discuss the assumptions during the discussion period.***
>
> ### W4: The role of structural results for maximal grouping function space in section 4.
>
> Section 4 connects to Section 5 by designing the function class used as input to the MC-Pseudolabel algorithm. **Together, Sections 4 and 5 establish a two-step paradigm for robust prediction**: first, a function class capturing priors of distribution shift is designed, then a downstream universal multi-calibration algorithm is run for any feasible grouping function class. In Section 4, the goal is to certify robustness by designing a grouping function class large enough to include the target density ratio while ensuring a feasible solution without exceeding the maximal grouping function space (L180-187). ***We would be happy to elaborate on the theorems' implications in Section 4 during the discussion period.***
>
> ### Q2, Q3: Edits for two sentences.
>
> Thanks for the suggestions on L83 and L18. Your interpretation is correct.
>
> ### Q4: Connection of extended multi-calibration definition with existing definitions.
>
> The major difference between Definition 2.1 and existing definitions is the joint grouping functions. The closest covariate-based version is in Globus-Harris et al. We also consider predictors with a continuous range, unlike their discrete outputs.
>
> ### Q5: Comparing Theorem 3.1 and Blasiok et al.
>
> Theorem 3.1 and its proof are essentially distinct from those of Blasiok et al. Blasiok et al. show a connection between smooth calibration error and post-processing gap. We follow a similar technique **only** in the last proof step for Theorem 3.1 (L740-741) to show a connection between a variation of their smooth calibration error and the post-processing gap. The rest of the proof shares no similarity with Blasiok et al. As a proof sketch, we first connect the multi-calibration error in the source distribution with the calibration error in each target distribution, then prove an equivalence between calibration error and a variation of smooth calibration error.
>
> ### Q6: Why do richer grouping function classes induce stronger generalizability?
>
> According to Theorem 3.4, a richer grouping function class implies generalizability to **more** distributions, as characterized by the density ratios in the function class. However, a predictor that is multi-calibrated to the function class may not exist, which is addressed in section 4.

---

> > ### Comment · Reviewer_ezjY · 2024-08-07
> >
> > I acknowledge and thank the authors for the response. In hindsight, and taking into account other reviewers insights, I believe my given score (3: reject) to be overly pessimistic. In my review, I agree that the technical contribution is quite strong. I also believe that the experiments are impressive and very comprehensive for a mainly theoretical paper. I have therefore increased my score to a 5.
> >
> > I still believe that the paper has much room for improvement in terms of presentation. In particular, I spent quite a long time understanding the main theorems / propositions from the paper (mainly section 4). I would appreciate either informal statements of the theorems, or expanded discussion before and after theorem statements. At the moment, this discussion seems to occur in different sections and not immediately preceding or succeeding the statements.

---

### Official Review · Reviewer_HkL5 · 2024-07-08

**Soundness:** 4
**Presentation:** 4
**Contribution:** 4
**Rating:** 8
**Confidence:** 4

**Summary:**

The authors explore an extension of multicalibration which includes joint grouping functions; groups that depend on both x and y. They show that multicalibration confers robustness to distribution shift problems. The authors then develop an optimization framework that post-processes a model to multicalibrate it. Finally, empirical demonstration of the robustness properties are given through experimentation.

**Strengths:**

The authors provide compelling reasons why multicalibration should be considered in modeling and model evaluation frameworks. In doing so, they provide a calibration framework that confers robustness on post-processed models. The proposed method appears significantly easier to use than methods that leverage actor-critic, domain confusion, and other modeling tricks to achieve robustness.

Section 4 shows that the condition "the grouping function class must include all density ratios between target and source measures" is achievable in practice.

The paper is well written and easy to follow.

**Weaknesses:**

A few questions are given below. I have nothing else to add.

**Questions:**

Line 9 of Algorithm 1 is unclear to me. I had thought A finds low risk models f(x) on an empirical dataset {(x,y)}. What does it mean for a grouping function to have low risk on a dataset? What is risk here?

With regard to Algorithm 1, the authors state "The prediction of grouping functions rectify the uncalibrated model and serves as pseudolabels for model updates." Can this statement be expanded, and details be provided?

**Limitations:**

The authors have adequately addressed limitations.

---

> ### Author Rebuttal · Authors · 2024-08-06
>
> Thank you for your support\! We really appreciate it that the reviewer identifies a compelling reason for applying multi-calibration to algorithmic robustness, and highly evaluates the simplicity and efficiency of our approach, the feasibility of our assumption, as well as the organization of the paper. The reviewer requests more details about an intuitive explanation for part of our algorithm, which we are ready to address below.
>
> ### Q1: Intuition behind regression on the grouping function class.
>
> In the following, we elaborate on the intuition described in Lines 255-257. In L9 of Algorithm 1, we perform regression on level sets with grouping functions. Here, the risk is the same as fitting a model $f(x)$. Since this regression is performed for each level of the predictor, it’s actually regressing the outcome on both the predictor’s output and the value of the grouping functions. Intuitively, this step rectifies the predictor given the distributional information conveyed by the grouping functions.
> Consider a toy case of multi-environment learning where the distributions in each environment are significantly different from each other, such that one can almost infer which environment a sample is taken from by looking at the density ratio provided by the grouping function. Then, regression on the predictor’s output and the value of the grouping functions reveals how much can be improved by knowing the environment from which the sample is taken. If there is indeed much improvement, it implies that the predictor does not perform well for this specific environment. In other words, the predictor shows a bias on this environment, which makes it uncalibrated. The improved prediction by regression on the grouping function class then serves as a pseudo label. Updating the model towards this improved prediction de-biases the model and makes it more calibrated on this particular distribution. The procedure is repeated in the next iteration to find another distribution where the model is uncalibrated, continuing until convergence.
> The intuitive explanation above sketches the proof of certified multi-calibration for this algorithm in Theorem 5.1.

---

### Official Review · Reviewer_TrUb · 2024-07-12

**Soundness:** 3
**Presentation:** 2
**Contribution:** 3
**Rating:** 5
**Confidence:** 2

**Summary:**

The paper explores multicalibration in the context of concept shifts, theoretically demonstrating the equivalence of multicalibration and invariance while providing a structural analysis of multicalibration. It introduces a novel algorithm that simplifies model selection and improves performance on real-world datasets.

**Strengths:**

- The paper provides rigorous and solid theoretical work for its claims.
- MC-Pseudolabel algorithm offers a practical tool that appears to integrate well with their frameworks.

**Weaknesses:**

- The empirical work is limited and not that convincing.
- Although technical solid, some assumptions might be too strong to achieve in reality.
- The complexity and detailed mathematical theories/assumptions would make the paper challenging for a broader audience to easily digest.

**Questions:**

- It seems that the setting you called "out-of-distribution generalization beyond covariate shift" is basically concept drift. I wonder why you choose to call that.
- In theorem 4.2 and 4.3, you consider an "absolutely continuous probability measure" without stating what measure it is with respect to.

**Limitations:**

See weakness.

---

> ### Author Rebuttal · Authors · 2024-08-06
>
> We appreciate the reviewer’s acknowledgement of our paper as both a solid theoretical work and a practical tool. Since the reviewer raises questions regarding the scope of experiments and the feasibility of assumptions, we are ready to offer more details for the reviewer’s evaluation.
>
> ### W1: Scope of experiments.
>
> In our paper, we have strived to conduct extensive experiments to cover a lot of ground, including multiple OOD learning settings, four different datasets across tabular and image modalities, various architectures of neural network predictive models, and multiple baselines both within and beyond the scope of invariant risk minimization. In detail, our experiments are performed on two settings of multi-environment learning with/wo environment annotation. We have a simulation dataset and three real datasets. Notably, PovertyMap comes from WILDS, the standard benchmark for OOD generalization, and ACSIncome is a popular benchmark for both fairness and algorithmic robustness, which features natural concept shifts \[32\]. Our models span linear, MLP and Resnet architectures. Our baselines include techniques from invariance risk minimization and other OOD methods, such as DRO and sample reweighting. The SOTA method (C-Mixup) from the WILDS open benchmark is also included. For all evaluations, we systematically select hyperparameters and conduct multiple repeated experiments according to the standard protocol of DomainNet. The proposed method achieves the best results in 7 out of the 8 reported evaluations.
>
> ### W2: Assumptions of theories.
>
> The main assumption of our results is stated in Equation 3, which requires that the grouping function class must include all density ratios between target and source measures. This assumption is general enough to essentially capture various practical OOD settings. Three such settings are discussed in Section 4.2 and Appendix B, where the assumption is satisfied, and the corresponding design of the grouping function class is given. For example, the subpopulation shift describes a setting where the target distribution is a different weighted mixture of subpopulations. This issue is effectively addressed by constructing the function class through interpolation of density ratios of several subpopulations in the data (Equation 11).
>
> ### Q1: Edits of title: beyond covariate shift or under concept shift.
>
> Before our work, the existing connection of multi-calibration and distribution shift was restricted to covariate shift. By using the term “beyond covariate shift,” we aim to highlight the gap we fill in the literature by being the first to extend the notion of multi-calibration and establish further connection to concept shift.
>
> ### Q2: Reference measure of Theorem 4.2, 4.3.
>
> We consider the Lebesgue measure as the reference measure.

---

> > ### Comment · Reviewer_TrUb · 2024-08-13
> >
> > Thank you for the rebuttal. It addresses most of my concerns. I'll maintain my score for now.

---

### Official Review · Reviewer_7ztk · 2024-07-14

**Soundness:** 4
**Presentation:** 3
**Contribution:** 3
**Rating:** 8
**Confidence:** 4

**Summary:**

This work studies an extension of multicalibration to include grouping functions of both covariates and labels. They show that just as multicalibration with respect to covariate density functions guarantees robustness to covariate shift, that extended multicalibration can imply robustness to concept shift. They further show a connection between extended multicalibrated predictors and invariant predictors. They study necessary and sufficient conditions on (covariate-dependent) grouping functions to achieve approximately Bayes optimal predictions on all target distributions, and introduce a boosting style algorithm that converges to an extended multicalibrated predictor under certain distributional assumptions. The usefulness of this new algorithm is empirically validated on PovertyMap, ACSIncome, and VesselPower.

**Strengths:**

This is a very interesting work that as far as I know if the first to considering multicalibration with respect to grouping functions that depend on labels. This gives a new approach to handling concept shift, and a new perspective on invariant predictors. I'm personally looking forward to follow-up work.

I found the paper very well-structured and easy to follow.

**Weaknesses:**

There are some questions regarding the feasibility of the MC-PseudoLabel that I did not see addressed in the work that would be helpful in understanding its usefulness. First, there's a known correspondence between multicalibration with respect to a hypothesis class H and weak agnostic learning for H, which unfortunately limits the grouping function classes for which we can efficiently obtain multicalibrated predictors. Does a similar correspondence also hold for extended multicalibration, or is a stronger weak learning condition required (alongside distributional assumptions)? I'm also curious about the necessary assumptions on the data for convergence of MC-PseudoLabel. On what kinds of distributions will the algorithm fail to convergence for H_{2, \Phi}?

Typos/suggested edits:

Abstract

“in existence” -> “in the presence”

Section 1

“flexibly designed to incorporates” -> “incorporate”

“Simultaneoulsy producing invariant predictors” -> “produce”

“porverty estimation”

Section 2

“which is learned in the source distribution” -> “which are learned”

2.2

“simultaneous approaches” -> “simultaneously approach”

“We show that multicalibration notion w.r.t.” -> “We show that our/this multicalibration notion”


Section 5

M(Sigma) is defined in the Appendix, but not in Theorem 5.2 where is first appears

**Questions:**

Repeated from weaknesses:

1. Is there a correspondence between weak agnostic learning and extended multicalibration?

2. On what kinds of distributions will the algorithm fail to convergence for H_{2, \Phi}?

**Limitations:**

The authors largely addressed the limitations (with the exception of possible computational limitations mentioned in Questions).

---

> ### Author Rebuttal · Authors · 2024-08-06
>
> We really appreciate the reviewer for pointing out the significance of our result for being the first to consider a label-dependent grouping function in multi-calibration, as well as the soundness and organization of our paper\! The reviewer has raised questions over the learnability of the extended grouping function class, which we are ready to address below.
>
> ### Q1: Correspondence between weak agnostic learning and extended multi-calibration.
>
> It is possible to establish a correspondence between a weak condition of the grouping function class and extended multi-calibration under certain distributional assumptions. By considering grouping functions within a linear space (section 4), multi-calibration with respect to a finite but sufficiently large dimensional grouping function class can imply multi-calibration with respect to an infinite dimensional grouping function class, which resolves the learnability of extended multi-calibration. This relationship parallels the way weak agnostic learning addresses the learnability of original multi-calibration. This conclusion is derived from the generalization theory of invariant risk minimization \[2\] for a linear data generation model. Arjovsky et al \[2\] conclude that an invariant predictor obtained from $O(d)$ distributions (d for the dimension of covariates) can generalize to infinitely many distributions (respecting the invariance assumption) whose density functions are linearly independent. Since one distribution corresponds to a density ratio function in our grouping function class, we conclude that multi-calibration with respect to an $O(d)$-dimensional grouping function class can imply multi-calibration with respect to an infinite-dimensional grouping function class, under a specific linear model. It would be very interesting to explore in future work whether such weak learning conditions can be extended to more general cases.
>
> \[2\] Martín Arjovsky, Léon Bottou, Ishaan Gulrajani, and David Lopez-Paz. Invariant risk minimization.
>
> ### Q2: Any distribution where the algorithm fails to converge?
>
> The algorithm fails to converge for grouping functions that violate the distributional assumption in Theorem 4.2. For example (L258-261), the algorithm will output the initial model $f\_0$ for the grouping function $h(x,y)=y$, because the pseudo labels always coincide with true labels. There does not exist a predictor that is multi-calibrated to $h(x,y)=y$, as it indicates an unresolvable concept shift where the outcome can arbitrarily change.
> More generally, the algorithm does not converge if the pseudo labels always coincide with the true labels, which also happens if the cardinality of the outcome’s support is smaller than the dimension of the grouping function class. This is more frequent for classification tasks, which is stated as a limitation of our work.

---

> > ### Comment · Reviewer_7ztk · 2024-08-12
> >
> > Thanks to the authors for the helpful response to my questions! I will keep my score.

---

### Decision · Program_Chairs · 2024-09-25

**Decision:**

Accept (poster)

**Comment:**

The paper provides new insights on multicalibration in the context of concept shifts. The reviewers unanimously acknowledge the notable technical contribution of the paper, however, the presentation can be improved for the final version. I recommend accept.